# Achieving Communication-Efficient Policy Evaluation for Multi-Agent Reinforcement Learning: Local TD-Steps or Batching?

## Abstract

In many consensus-based actor-critic multi-agent reinforcement learning (MARL) strategies, one of the key components is the MARL policy evaluation (PE) problem, where a set of $N$ agents work cooperatively to evaluate the value function of the global states under a given policy only through communicating with their neighbors. In MARL-PE, a critical challenge is how to lower the communication complexity, which is defined as the rounds of communication between neighboring nodes in order to converge to some $\epsilon$-stationary point. To lower communication complexity in MARL-PE, there exist two "natural" ideas: i) using batching to reduce the variance of TD (temporal difference) errors, which in turn improves the convergence rate of MARL-PE; and ii) performing multiple local TD update steps between each consecutive rounds of communication, so as to reduce the communication frequency. While the effectiveness of the batching approach has been verified and relatively well-understood, the validity of the local TD-steps approach remains unclear due to the potential "agent-drift" phenomenon resulted from various heterogeneity factors across agents. This leads to an interesting open question in MARL-PE: *Does the local TD-steps approach really work and how does it perform in comparison to the batching approach?* In this paper, we take the first attempt to answer this fundamental question. Our theoretical analysis and experimental results confirm that allowing multiple local TD steps is indeed a valid approach in lowering the communication complexity of MARL-PE compared to vanilla consensus-based MARL-PE algorithms. Specifically, the local TD steps between two consecutive communication rounds can be as large as $\mathcal{O}(\sqrt{1/\epsilon}\log(1/\epsilon))$ in order to converge to an $\epsilon$-stationary point of MARL-PE. Theoretically, we show that in order to reach the optimal sample complexity up to a log factor, the communication complexity is $\mathcal{O}(\sqrt{1/\epsilon}\log(1/\epsilon))$, which is worse than TD learning with batching, whose communication complexity is $\mathcal{O}(\log(1/\epsilon))$. However, the experimental results show that the allowing multiple steps can be as good as the batch approach.

## 1 Introduction

1) **Background and Motivation:** With the recent success of reinforcement learning (RL) (Sutton & Barto, 2018) techniques in the dynamic decision making process where the underlying model is unkown, MARL, a natural extension of RL to multi-agent systems, has also found increasing applications. Compared to traditional RL, the richness of multi-agent systems has given rise to far more diverse problem settings in MARL, including cooperative, competitive, and mixed MARL (see (Zhang et al., 2021a) for an excellent survey). In this paper, we are interested in *cooperative* MARL, which has found a wide range of applications in the field of networked large-scale systems, such as power networks (Chen et al., 2022; Riedmiller et al., 2000), autonomous driving (Yu et al., 2019; Shalev-Shwartz et al., 2016) and so on. A defining feature of cooperative MARL is that all agents in the system collaborate to learn a joint policy to maximize long-term system-wide total reward through communicating with each other. However, due to the decentralized nature (i.e., lack of a centralized infrastructure) of cooperative MARL, the collaboration between the agents can only rely on some algorithmic designs to induce a "consensus" that can be reached by the agents.

In many consensus-based actor-critic MARL strategies, one of the key components is the MARL policy evaluation (PE) problem, where a set of $N$ agents work cooperatively to evaluate the value function of the global states for a given joint policy. Just as in the PE problem of conventional RL, temporal difference (TD) learning (Sutton, 1988) has been the prevailing method for MARL-PE thanks to its simplicity and empirical successes in real-world applications. Simply speaking, the key idea of TD learning is to learn the value function by using the Bellman equation to bootstrap from the current estimated value function. However, as mentioned earlier, the decentralized nature of the MARL-PE problem necessitates communication among agents for TD learning. Hence, a critical challenge in consensus-based MARL-PE is how to lower the communication complexity, which is defined as the required rounds of communication between neighboring agents in order to converge to some $\epsilon$-stationary point of the MARL-PE problem.

To lower communication complexity for solving MARL-PE problems, there exist two "natural" ideas: i) using batching of trajectory samples to reduce the variance of TD errors, which in turn improves the convergence rate of MARL-PE; and ii) using an "infrequent communication" approach where we perform multiple local TD update steps between each consecutive rounds of communication to reduce the communication frequency. While the effectiveness of the "batching" approach has been verified and relatively well-understood (Hairi et al., 2022; Chen et al., 2021), the validity of the "local TD-step" approach remains unclear due to the potential "agent-drift" phenomenon resulted from various heterogeneity factors across agents (more on this soon). This leads to two interesting open questions: *1) Can the local TD-steps approach really lower the communication complexity for solving MARL-PE? 2) If the answer to 1) is "yes," how does the local TD-steps approach perform in comparison to the batching approach?* Answering these two questions from both theory and in practice constitutes the main goal of this paper.

2) **Technical Challenges:** Answering the above questions is highly non-trivial due to several technical challenges in the convergence analysis of the local TD-steps approach. Notably, it is easy to see that the structure of TD learning in consensus-based cooperative MARL resembles that of decentralized stochastic gradient descent (DSGD) method in consensus-based decentralized optimization(Nedic & Ozdaglar, 2009; Lian et al., 2017; Pu & Nedić, 2021). Thus, it is tempting to believe that one can borrow convergence analysis techniques of DSGD and apply them in TD learning. However, despite such similarities, there also exist significant differences between TD learning in MARL and DSGD.

- *Structural Differences:* First, we note that TD learning is *not* a true gradient-based method, since TD error is not a gradient estimator of any static objective function as in DSGD, which is well defined in a consensus-based decentralized optimization problem. Also, in decentralized optimization, the gradient terms are often assumed to be bounded. However, in TD learning, the TD-errors can not be assumed to be bounded without further assuming that the value function approximation parameters lie in some compact set.

- *Markovian Data in TD Learning:* In RL/MARL problem, there exists an underlying Markovian dynamic process across time steps, where the state distribution may differ at different time steps. By contrast, in decentralized optimization, it is often safe to assume that the data at each agent are independently distributed. Thus, it is not applicable to directly use convergence analysis techniques of decentralized optimization in TD learning for MARL-PE. The coupling and dependence among samples renders the convergence analysis of TD learning in MARL far more challenging.

- *"Agent-Drift" Phenomenon:* Due to heterogeneity nature of the rewards across agents, executing multiple local TD update steps would inevitably pull the local function approximation parameters toward the direction of approximating local value functions rather than the global value function, leading to the "agent-drift" phenomenon. Hence, it is unclear whether such local steps help or hurt the convergence of TD learning in MARL. Intuitively, if the agent-drift is too large, then the low communication-complexity benefit of infrequent communication might be offset by the errors in aligning the local model parameters at each agent to the true global value function parameters. Because of the agent-drift effect, the number of local TD update steps has to be chosen judiciously to mitigate the potentially large divergence of the function approximation parameters between each communication round.

3) **Main Results and Contribution:** The main contribution of this paper is that we overcome the above challenges in analyzing the communication complexity of the local TD-steps approach for cooperative MARL-PE. By doing so, we are able to shed light on the feasibility and effect of local

TD steps in the consensus-based TD learning in the MARL-PE problem. We summarize our main results in this paper as follows:

- Both theoretically and empirically, we show that allowing multiple local TD steps is indeed a valid approach that can significantly lower the communication complexity of MARL-PE compared to vanilla consensus-based decentralized TD learning algorithms Doan et al. (2021; 2019); Zhang et al. (2018). Specifically, we show in theory that to under the condition of achieving $\mathcal{O}(1/\epsilon \log^2(1/\epsilon))$ sample complexity (which differs from the state-of-the-art sample complexity only by a log factor), the local TD-steps approach can allow up to $\mathcal{O}(\sqrt{1/\epsilon} \log(1/\epsilon))$ local steps and the communication complexity is $\mathcal{O}(\sqrt{1/\epsilon} \log(1/\epsilon))$. Compared to vanilla algorithms, this improves the communication complexity by a factor of $\mathcal{O}(\sqrt{1/\epsilon})$.

- Moreover, we show in theory that although executing multiple local TD steps improves the communication complexity significantly, its improvement may *not* be as good as that of the batch approach under the same sample complexity condition Hairi et al. (2022); Chen et al. (2021). Specifically, by choosing the batch size to be $\mathcal{O}(1/\epsilon)$, the communication complexity of the batching approach can achieve $\mathcal{O}(\log(1/\epsilon))$. This indicates that the local TD-steps approach "helps," but may not as much as the batch approach.

- Although the communication complexity bound is relatively worse, the extensive empirical results show that the performance of the local TD approach can be comparable to that of batching approach both in synthetic and real-world setting.

The rest of the paper is organized as follows. In Section 2, we review the literature to put our work in comparative perspectives. In Section 3, we present the system model and formulation of the MARL-PE problem. In Section 4, we introduce the decentralized TD learning algorithm with multiple local TD steps for MARL-PE. In Section 5, we provide the theoretical convergence analysis for the decentralized TD learning algorithm with multiple local TD steps. Section 6 presents numerical results and Section 7 concludes this paper. Due to space limitation, some proof details are relegated to the supplementary material.

## 2 RELATED WORK

In this section, we provide an overview on two lines of research that are related to this work: i) MARL-PE and ii) single-agent RL policy evaluation.

**1) Multi-agent reinforcement learning policy evaluation:** To our knowledge, the work in (Zhang et al., 2018) proposed the first fully decentralized multi-agent actor-critic algorithm using TD learning in the critic step, which solves the PE problem. However, the convergence results for both its critic and actor steps are asymptotic. Finite-time analysis of MARL-PE problem using distributed TD learning algorithm has been first studied in (Doan et al., 2019) under the i.i.d. sampling assumption, later the work in (Doan et al., 2021) generalized the result to Markovian sampling assumption. In (Lin et al., 2019), a compressed algorithm is proposed where, instead of sending a vector, only a single entry is sent during communication is proposed. However, their communication complexity (i.e., the number of communication rounds) remains the same as sample complexity and the convergence is only asymptotic. In (Chen et al., 2018), a lazy communication algorithm is proposed assuming a central controller, which is different from the fully decentralized setting we consider in this paper. There exists another class of approaches (Zhang et al., 2021b; Macua et al., 2014; Lee et al., 2018; Wai et al., 2018; Ren & Haupt, 2019) that solve MARL-PE problem by formulating it into optimizing projected Bellman error or its variants, where the proposed algorithms require frequent communication. This class of algorithms do not use the on-policy TD learning approach as we do in our paper. In (Kim et al., 2019), the paper optimizes communication in order to comply the bandwidth restriction and minimize the collision between pair-wise channels. However, the work adopts centralized learning and distributed execution paradigm, where in our paper, the learning process is fully decentralized.

However, most of the existing distributed TD learning algorithms (Zhang et al., 2018; Doan et al., 2019; 2021) for MARL perform frequent consensus rounds (i.e., one round of communication per local TD update) to share the approximation parameters among neighbors. Specifically, in these algorithms, agents share the parameters to their neighbors in every time step, which causes the

communication complexity to be the same as the sample complexity. In this paper, we consider an infrequent communication framework that allows the agents to do multiple local TD learning steps and share the value function approximation parameters one every $K(>1)$ rounds. In (Hairi et al., 2022; Chen et al., 2021), complete actor-critic algorithms have been proposed and batching approach has been used in the critic step, which corresponds to MARL-PE. In such batch approach, consensus is performed in every $M = \infty/\epsilon$ batch samples, which in turn only requires $O(\log 1/\epsilon)$ communication complexity. To investigate the effect of such local steps solely, we do not consider further modifications of batching as in (Xu et al., 2020; Hairi et al., 2022; Chen et al., 2021).

**2) Single-agent reinforcement learning policy evaluation:** For single-agent RL, policy evaluation problems have been extensively studied in terms of asymptotic convergence (Tsitsiklis & Van Roy, 1997; 1999; 2002), finite-time convergence under i.i.d. sampling assumption Lakshminarayanan & Szepesvari (2018) and under Markovian sampling assumption using different techniques (Srikant & Ying, 2019; Bhandari et al., 2018). Further, using batched TD learning (Xu et al., 2020) yields state-of-the-art sample complexity $O((1/\epsilon)\log(1/\epsilon))$. However, there is no notion of "communication with other agents" due to the single-agent nature. Thus, results in this area are not directly comparable to our work.

# 3 DISTRIBUTED POLICY EVALUATION IN MULTI-AGENT REINFORCEMENT LEARNING

Throughout this paper, $\|\cdot\|$ denotes the $\ell_2$-norm for vectors and the $\ell_2$-induced norm for matrices. $\|\cdot\|_F$ denotes the Frobenius norm for matrices. $(\cdot)^T$ denotes the transpose for a matrix or a vector.

## 3.1 SYSTEM MODEL

Consider a multi-agent system with $N$ agents, denoted by $\mathcal{N} = \{1, \cdots, N\}$, operating in a networked environment. Let $\mathcal{E}$ be the edge set for a given network $\mathcal{G} = (\mathcal{N}, \mathcal{E})$. To formulate our MARL problem and facilitate our subsequent discussions, we first define the notion of networked multi-agent Markov decision process (MDP) as follows.

**Definition 1** (Networked Multi-Agent MDP). Let $\mathcal{G} = (\mathcal{N}, \mathcal{E})$ be a communication network that connects $N$ agents. A networked multi-agent MDP is defined by following six-tuple $(\mathcal{S}, \{\mathcal{A}^i\}_{i \in \mathcal{N}}, P, \{R^i\}_{i \in \mathcal{N}}, \mathcal{G}, \gamma)$, where $\mathcal{S}$ is the global state space observed by all agents, $\mathcal{A}^i$ is the action set for agent $i$, $P : \mathcal{S} \times \mathcal{A} \times \mathcal{S} \to [0, 1]$ is a global state transition function, $R^i : \mathcal{S} \times \mathcal{A}$ is the local reward function for agent $i$, and $\gamma \in (0, 1)$ is the discount factor. Let $\mathcal{A} = \prod_{i \in \mathcal{N}} \mathcal{A}^i$ be the joint action set of all agents.

In this paper, we assume that the global state space $\mathcal{S}$ is finite. We also assume that at time step $t \geq 0$, all agents can observe the current global state $s_t$. However, each agent can only observe its own reward $r_t^i$, i.e., agents do not observe or share rewards with other agents at time $t$. The reward function $R^i(s, a)$ is an expectation given $s$ and $a$, and the instantaneous reward is denoted by $r^i(s, a)$, i.e., $R^i(s, a) = \mathbb{E}[r^i(s, a)]$.

We consider policies that are stationary. In our MARL system, each agent chooses its action following its local policy $\pi^i$ that is conditioned on the current global state $s$, i.e., $\pi^i(a^i|s)$ is the probability for agent $i$ to choose an action $a^i \in \mathcal{A}^i$. Then, the joint policy $\pi : \mathcal{S} \times \mathcal{A} \to [0, 1]$ can be written as $\pi(a|s) = \prod_{i \in \mathcal{N}} \pi^i(a^i|s)$.

The global value function for all $s \in \mathcal{S}$ is defined as follows:

$$V(s) = \mathbb{E}\left[\sum_{t=0}^{\infty} \frac{\gamma^t}{N} \sum_{i \in \mathcal{N}} r^i(s_t, a_t) \Big| s_0 = s\right].$$

## 3.2 TECHNICAL ASSUMPTIONS

We now state the following assumptions for the MARL system described above.

**Assumption 1.** For the given policy $\pi$, we assume the induced Markov chain $\{s_t\}_{t \geq 0}$ is irreducible and aperiodic.

**Assumption 2.** The instantaneous reward $r_t^i$ is uniformly bounded by a constant $r_{\max} > 0$ for any $i \in \mathcal{N}$ and $t \geq 0$.

**Assumption 3.** Let $A$ be a consensus weight matrix for a given communication network $\mathcal{G}$. There exists a positive constant $\eta > 0$ such that $A \in \mathbb{R}^{N \times N}$ is doubly stochastic and $\boldsymbol{A}_{ii} \geq \eta, \forall i \in \mathcal{N}$. Moreover, $\boldsymbol{A}_{ij} \geq \eta$ if $i, j$ are connected, otherwise $\boldsymbol{A}_{ij} = 0$.

**Assumption 4.** The global value function is parameterized by linear functions, i.e., $V(s; w) = \phi(s)^T w$ where $\phi(s) = [\phi_1(s), \cdots, \phi_d(s)]^T \in \mathbb{R}^d$ is the feature associated with the state $s \in \mathcal{S}$ and $d < |\mathcal{S}|$. The feature vectors $\phi(s)$ are uniformly bounded for any $s \in \mathcal{S}$. Without loss of generality, we assume that $\|\phi(s)\| \leq 1$. Furthermore, the feature matrix $\Phi \in \mathbb{R}^{|\mathcal{S}| \times d}$ is full column rank.

Assumption 1 guarantees that there exists a unique stationary distribution over $\mathcal{S}$ for the induced Markov chain by the given policy $\pi$. Assumption 2 is common in the RL literature (see, e.g., (Zhang et al., 2018; Xu et al., 2020; Doan et al., 2019)) and easy to be satisfied in many practical MDP models with finite state and action spaces. Assumption 3 is standard in the distributed multi-agent optimization literature (Nedic & Ozdaglar, 2009). This assumption says that non-zero entries of the weight matrix $A$ needs to be lower bounded by a positive value $\eta$. Assumption 4 on features is standard and has been widely adopted in the literature, e.g., (Tsitsiklis & Van Roy, 1999; Zhang et al., 2018; Qiu et al., 2021; Srikant & Ying, 2019). The goal of this assumption is to approximate the value function as follows:

$$V(s) \approx V(s; w) = \phi(s)^T w$$

where $\phi(s)$ is the feature associated with state $s \in \mathcal{S}$. As a result, $\nabla_w V(s; w) = \phi(s)$ for all $s \in \mathcal{S}$.

## 4 DECENTRALIZED TD LEARNING WITH LOCAL STEPS FOR MARL

In this section, we introduce the decentralized TD learning algorithm with local TD steps (i.e., infrequent communication), which is illustrated in Algorithm 1. Given a policy $\pi$, the goal of the MARL-PE in the decentralized setting is to collaboratively characterize the value function. Specifically, each agent $i$ maintains a value function approximation parameter $w^i$ locally, which estimates the global value function as follows:

$$V(s; w^i) = \phi(s)^T w^i.$$

Our algorithm is designed with two loops. The outer loop is the communication rounds, where consensus update (Line 11 in Algorithm 1) is performed for $L$ rounds. The inner loop is local TD update steps, which consists of $K$ steps. Locally, each agent performs local TD learning updates within each communication round $0 \leq l \leq L - 1$ as follows:

$$w_{l,k+1}^i = w_{l,k}^i + \beta \cdot \delta_{l,k}^i \cdot \phi(s_{l,k}), \tag{1}$$

where $\beta > 0$ is the step size and $\delta_{l,k}^i$ is the local TD error, which is defined as follows

$$\delta_{l,k}^i := r_{l,k+1}^i + \gamma \phi(s_{l,k+1}) w_{l,k}^i - \phi(s_{l,k}) w_{l,k}^i.$$

We note that equation 1 is considered one local TD learning step. Within each inner loop, this local TD update step is done $K$ times.

Due to the privacy of the reward signals in the fully decentralized setting, the agents are unable to access the rewards of any other agents, let alone the average rewards. Therefore, communication/sharing of the value function approximation parameters among the neighbors is necessary (Zhang et al., 2018; Doan et al., 2019; Chen et al., 2021; Hairi et al., 2022). This step is also referred to as consensus update, which can be typically done as follows:

$$w_{l+1,0}^i = \sum_{j \in \mathcal{N}_i} A_{ij} w_{l,K}^i. \tag{2}$$

In other words, after performing $K$ local TD steps, each agent shares its parameter to the neighbors, receives the ones from the neighbors, and then updates its own parameter in a weighted aggregation as shown in equation 2. We note that in our algorithm, the infrequent communication is done by agents communicating only periodically with the period being $K$. The decentralized TD learning with periodic local steps is concluded in Algorithm 1. We also note that when $K = 1$, our algorithm reduces to the vanilla distributed TD learning algorithm (Doan et al., 2019; 2021; Zhang et al., 2018). Hence, the vanilla distributed TD learning can be viewed as a special case of our algorithm.

---

**Algorithm 1:** Decentralized TD Learning with periodic local steps

---

**Input** : Initial state $s_0$, $\pi = \{\pi^i | i \in \mathcal{N}\}$, feature map $\phi$, initial parameters $\{w_{0,0}^i | i \in \mathcal{N}\}$, step size $\beta$, communication round number $L$, local step number $K$

1 **for** $l = 0, \cdots, L-1$ **do**
2     **for** $k = 0, \cdots, K-1$ **do**
3        $s_{l,0} = s_{l-1,K}$ (when $l = 0$ and $k = 0$, $s_{l,k} = s_0$);
4        **for** *all* $i \in \mathcal{N}$ **do**
5           Execute action $a_{l,k}^i \sim \pi^i(\cdot | s_{l,k})$;
6           Observe the state $s_{l,k+1}$ and reward $r_{l,k+1}^i$;
7           Update $\delta_{l,k}^i \leftarrow r_{l,k+1}^i + \gamma \phi(s_{l,k+1})^T w_{l,k}^i - \phi(s_{l,k})^T w_{l,k}^i$;
8        **end**
9        Local TD Step: $w_{l,k+1}^i \leftarrow w_{l,k}^i + \beta \delta_{l,k}^i \cdot \phi(s_{l,k})$;
10     **end**
11     Consensus Update: $w_{l+1,0}^i \leftarrow \sum_{j \in \mathcal{N}_i} A(i,j) \cdot w_{l,K}^j$;
12 **end**
   **Output:** $w_{L,0}$

---

## 5   CONVERGENCE ANALYSIS OF DECENTRALIZED TD LEARNING WITH LOCAL STEPS

In this section, we present the convergence results for Algorithm 1, which further imply both the sample and communication complexities of Algorithm 1.

To characterize the convergence, we define the following quantities:

$$\Psi := \mathbb{E}[(\gamma\phi(s') - \phi(s))\phi^T(s)] \qquad \text{and} \qquad b := \frac{1}{N}\mathbb{E}[\phi(s)\sum_{i \in \mathcal{N}} r^i(s,a)]. \qquad (3)$$

The expectations in equation 3 are taken over the steady state distribution induced by the policy, which is guaranteed to exist due to Assumption 1, stationary action policy $a \sim \pi(\cdot|s)$, state transition probability $s' \sim P(\cdot|s,a)$, and the reward distributions. Furthermore, we define

$$w^* = \Psi^{-1}b, \qquad (4)$$

where the invertibility is due to $\Psi$ being negative definite (Tsitsiklis & Van Roy, 1997). Consequently, we define mixing time $\tau(\beta)$ as follows

$$\|\Psi - \mathbb{E}[(\gamma\phi(s_{k+1}) - \phi(s_k))\phi^T(s)|s_0 = s]\| \leq \beta \quad \forall s, \forall k \geq \tau(\beta) \qquad (5a)$$

$$\left\|b - \frac{1}{N}\mathbb{E}[\phi(s_k)\sum_{i \in \mathcal{N}} r^i(s_k, a_k)|s_0 = s]\right\| \leq \beta \quad \forall s, \forall k \geq \tau(\beta), \qquad (5b)$$

where the expectation is state distribution at the $k$-th time step, stationary action policy $a_k \sim \pi(\cdot|s_k)$, state transition probability $s_{k+1} \sim P(\cdot|s_k, a_k)$ and reward distribution. We note that under the Assumption 1, by (Levin & Peres, 2017, Theorem 4.9), the Markov chain mixes at a geometric rate, which implies $\tau(\beta) = \mathcal{O}(\log \frac{1}{\beta})$.

The choice of step size $\beta$ on the right-hand-side (RHS) of the definition of mixing time in equation 5 is for simplicity. From (Srikant & Ying, 2019), we know that the error level at the steady state is order-wise proportional to the step size up to a log factor (see the second term in the RHS of Eq. (12) of Srikant & Ying (2019)), which is due to the geometric mixing time property.

Before presenting our main theorem, we introduce two useful lemmas. Our analysis is to divide the convergence error into two parts, the consensus error, which is defined as the agent's parameters deviation from the average parameter, and convergence error of the average parameter to the solution of the ODE in equation 4.

First, we define the average of the parameters to be $\bar{w}_{l,k} = \frac{1}{N}\sum_{i \in \mathcal{N}} w_{l,k}^i$ for any communication round $0 \leq l \leq L-1$ and local step $0 \leq k \leq K-1$. Then, we define the consensus error as:

$$Q_{l,k}^i := w_{l,k}^i - \bar{w}_{l,k} \qquad (6)$$

and the matrix form is $Q_{l,k} = [Q_{l,k}^1, \cdots, Q_{l,k}^N] \in \mathbb{R}^{d \times N}$.

We provide an upper bound for the consensus error generated by Algorithm 1 in the following lemma.

**Lemma 1.** *Suppose that Assumptions 2–4 hold. For the consensus error generated by Algorithm 1, if $\beta K \leq \min\{\frac{1}{2}, \frac{\eta^{N-1}}{4(1-\eta^{N-1})}\}$, it holds that*

$$\|Q_{L,0}\| \leq \kappa_1 \rho^L \|Q_{0,0}\| + \frac{\kappa_2 \beta K}{1 - \rho}, \tag{7}$$

*where $\kappa_1 = \frac{2N^2(1+\eta^{-(N-1)})}{1-\eta^{N-1}}$, $\kappa_2 = 4(1 + \eta^{-(N-1)})N^{\frac{5}{2}} r_{\max}$ and $\rho := (1 + 4\beta K)(1 - \eta^{N-1})$. By the condition on $\beta K$, we have $0 < \rho < 1$.*

Lemma 1 shows that even if the parameters are not set to be the same initially, the effect of the initial consensus error will converge to $0$ exponentially fast as the round of communication goes to infinity. The second term is linear with respect to $\beta K$, which resembles the constant term in optimization using constant step size SGD. This product term will dictates the consensus error and the error level that the algorithm converges to, see Section A.3 for detail. The proof for the lemma is relegated to Section C.2.

Next, we provide a lemma that characterizes the convergence of the average parameter $\bar{w}_{l,k}$. Note that, here we utilize the Theorem 7 in (Srikant & Ying, 2019), which has the state-of-the-art non-batch sample complexity for TD learning.

**Lemma 2.** *Suppose Assumptions 1-4 hold. For the parameters generated by Algorithm 1, we have following result for the average of them*

$$\mathbb{E}[\|\bar{w}_{L,0} - w^*\|^2] \leq c_2(1 - c_1\beta)^{KL - \tau(\beta)}(\|\bar{w}_{0,0} - w^*\| + \frac{r_{\max}}{3})^2 + c_3 \beta \tau(\beta), \tag{8}$$

*where $c_1, c_2, c_3 > 0$ are constants that are independent of step size $\beta$, local steps $K$ and communication round $L$ and $\tau(\beta)$ is the mixing time. See the specified expressions of the constants and discussion in Section C.3.*

The average parameter $\bar{w}_{L,0} = \frac{1}{N} \sum_{i \in \mathcal{N}} w_{L,0}^i$ corresponds to the updates after $K \times L$ iterations (samples). Lemma 2 shows that the average of the parameters converges to solution of the ODE with the rate given by the RHS of equation 8.

Now, we state the main convergence result of Algorithm 1:

**Theorem 1.** *Suppose that Assumptions 1-4 hold. For the given policy, consider the iteration generated by Algorithm 1. If $\beta K \leq \min\{\frac{1}{2}, \frac{\eta^{N-1}}{4(1-\eta^{N-1})}\}$, it then follows that:*

$$\mathbb{E}\left[\sum_{i=1}^N \|w_{L,0}^i - w^*\|^2\right] \leq 2d\left(\kappa_1 \rho^L \|Q_{0,0}\| + \frac{\kappa_2 \beta K}{1 - \rho}\right)^2$$
$$+ 2N\left(c_2(1 - c_1\beta)^{KL - \tau}(\|\bar{w}_{0,0} - w^*\| + \frac{r_{\max}}{3})^2 + c_3 \beta \tau\right), \tag{9}$$

*where $\kappa_1, \kappa_2, c_1, c_2, c_3 > 0, 0 < \rho < 1$ are constants, and $\bar{w}_{0,0} = \frac{1}{N} \sum_{i \in \mathcal{N}} w_{0,0}^i$ and $Q_{0,0}$ is the initial consensus error defined in equation 6. Furthermore, by letting $\beta = \Theta(\epsilon \log^{-1}(1/\epsilon))$, $K = \Theta(\sqrt{1/\epsilon} \log(1/\epsilon))$ and $L = \Theta(\sqrt{1/\epsilon} \log(1/\epsilon))$, we have $\mathbb{E}[\sum_{i=1}^N \|w_{L,0}^i - w^*\|^2] = \mathcal{O}(\epsilon)$. The sample complexity is $KL = \mathcal{O}(1/\epsilon \log^2(1/\epsilon))$ and the communication complexity is $L = \mathcal{O}(\sqrt{1/\epsilon} \log(1/\epsilon))$.*

Note that since we use double loops in Algorithm 1, the parameter $w_{L,0}^i$ corresponds to the iteration result after $K \times L$ samples (iterations). We remark that the sample complexity of $\mathcal{O}((1/\epsilon) \log^2(1/\epsilon))$ is the state-of-the-art sample complexity for non-batching single agent RL policy evaluation problem Srikant & Ying (2019). The sample complexity of our algorithm in distributed multi-agent setting, which is also a non-batching approach, matches this sample complexity. In addition, compared with single-agent policy evaluation (Xu et al., 2020), which is a batching method and its multi-agent

counterpart (Hairi et al., 2022; Chen et al., 2021), this sample complexity only differs by a $\log$ factor. We note that, in (Xu et al., 2020; Hairi et al., 2022; Chen et al., 2021), the algorithms are complete actor-critic algorithms. Thus, we only compare our results with their critic steps, which solve the policy evaluation problem.

In our proposed algorithm, between consecutive communication rounds, the number of local TD steps for each agent can be $K = \mathcal{O}(\sqrt{1/\epsilon}\log(1/\epsilon))$. This improved the communication complexity of vanilla distributed TD algorithms (Zhang et al., 2018; Doan et al., 2019; 2021) by a factor of $K = \mathcal{O}(\sqrt{1/\epsilon}\log(1/\epsilon))$. The communication complexity of our algorithm is $L = \mathcal{O}(\sqrt{1/\epsilon}\log(1/\epsilon))$, which, interestingly, equals to $K$ order-wise.

## 6 EXPERIMENTAL RESULTS

In this section, we conduct numerical experiments to compare our proposed local TD-update algorithm, TD learning with local steps, with vanilla TD learning (Zhang et al., 2018; Doan et al., 2019; 2021) and the batch TD learning (Hairi et al., 2022; Chen et al., 2021) in both synthetic settings as in (Zhang et al., 2018) and cooperative navigation tasks as in (Lowe et al., 2017). Due to the page limit, we relegate the experiment setups and rigorous definitions of the performance metrics in the Appendix A.1 and B.1.

### 6.1 CONVERGENCE PERFORMANCE ON SYNTHETIC SETTING

In the synthetic experiment, we show that the function approximation parameter in the proposed local TD algorithm converges and the convergence with respect to the number of communication round for all algorithms. In Figure 1, the y-axis is the normalized convergence error of the left hand side of equation 9 and the axes are the numbers of communication rounds in Figure 1a,1c and sample numbers in Figure 1b, 1d. For extensive experimental details, discussions and results of experiment setup, convergence performance, the impact of local steps and linear convergences, please see Appendix A.

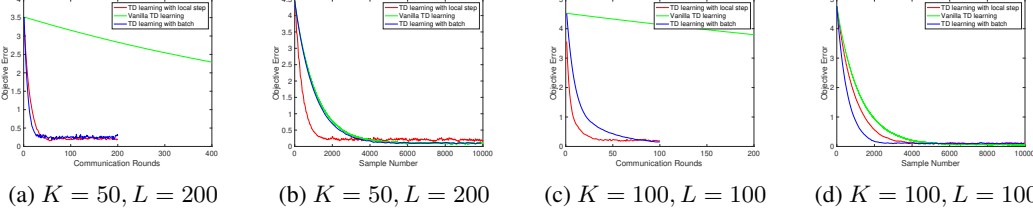

(a) $K = 50, L = 200$    (b) $K = 50, L = 200$    (c) $K = 100, L = 100$    (d) $K = 100, L = 100$

Figure 1: Convergence with respect to Communication Rounds and Sample Number

We considered 20 agents over a ring network. In Figure 1a, we provide the convergence results with respect to the communication rounds for all three algorithms, where the local step $K = 50$ for local TD algorithm and the batch size is 50 for batch algorithm. Under such a setting, both local TD algorithm and batch TD algorithm only performs consensus communication every 50 samples. To keep the comparison fair, we keep the local step number and batch size to be the same for the majority of the comparisons except Figure 4. We can see that within 200 communication rounds, both batch algorithm and local TD algorithm converge to a very similar level, yet vanilla TD algorithm doesn't converge even after 400 rounds of communication. Between the local TD algorithm and batch algorithm, the batch algorithm converges slightly faster, which means requiring slightly fewer rounds of communication among the agents. In Figure 1c when local step $K = 100$ for local TD algorithm and batch size is 100 for batch algorithm, the local TD algorithm requires the least amount of communication rounds to converge, even compared to batch algorithm. On the other hand, local TD algorithm performs significantly better compared to vanilla algorithm. Note that in these parameter settings, our algorithm has the same number of communication rounds as the batched TD algorithm. In Figures 1b and 1d, we present the corresponding convergence results with respect to sample numbers. We can see that vanilla TD eventually converges but requires consensus operation after each sampling. For more detailed discussion on the convergence with respect to sample numbers, please see Section A.2 and A.4.

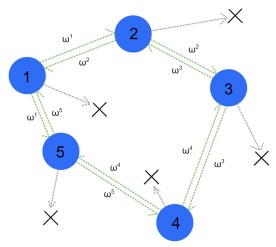 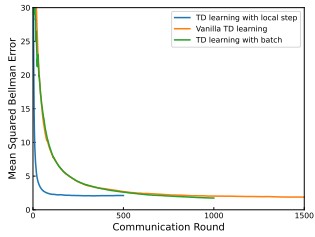 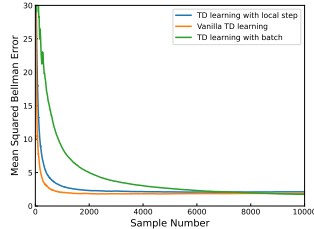

(a) Cooperative Navigation Task

(b) Bellman Error over Communication Rounds

(c) Bellman Error over Sample Number

Figure 2: Convergence with respect to Communication Rounds and Sample Number

Figure 1 verifies the theoretical analysis that allowing local steps does help with the communication rounds compared to vanilla algorithm. In addition, even though the theoretical sample complexity of local TD algorithm is worse than batch algorithm, in practice, the communication rounds are comparable between these two approaches. This may be because the sample complexity is derived from the upper bound of the finite time analysis in Theorem 1 and this upper bound is not only an order-wise result with respect to some $\epsilon$ criteria, the tightness of the upper bounds depends on the analysis techniques. This may indicate that there is room to improve for the finite time analysis of the local TD algorithm.

### 6.2 CONVERGENCE PERFORMANCE ON COOPERATIVE NAVIGATION TASK

In the cooperative navigation task, the agents (blue circles) are trained to cover the landmarks (crosses) as illustrated in Figure 2a. Agents observe positions and velocities of all agents and collaboratively cover the landmarks while avoiding collisions. The rewards for agents are defined through the proximity to the nearest landmarks. Unlike the synthetic experiment, the fixed point of the corresponding ODE as in equation 4 can't be calculated, so we use the squared Bellman error as the performance metric. For extensive simulation results, which include discussions on various network topologies, local steps/batch sizes, different step sizes, and consensus error metrics, please see Appendix B.

For the results in Figure 2b and 2c, we consider $N = 9$ agents over an Erdos-Renyi (ER) network and set local step $K$ to be 20 for local TD algorithm and batch size to be 10 for batch algorithm and appropriate step sizes for the algorithms. Similar to the synthetic experiment, we can see that all algorithms converge to a very similar Bellman error. This again verifies the theoretical analysis that allowing local steps and performing infrequent communications is feasible and can converge. Moreover, in this setting, our local TD algorithm converges much faster in terms of the communication round. Specifically, local TD algorithm requires much less than 500 rounds of communication to converge, and both batch algorithm and vanilla algorithm perform similarly and require more than 500 rounds of communication to converge.

## 7 CONCLUSION

In this paper, we answer the questions of whether local TD steps can lower the communication complexity of distributed TD learning and, if so, how is the performance in comparison with batch TD learning. Our theoretical analysis and experimental results show that the local TD steps can significantly lower the communication complexity compared to vanilla TD learning. In addition, our theoretical analysis also shows that the local step can be as large as $K = \mathcal{O}(\sqrt{1/\epsilon}\log(1/\epsilon))$ to converge to $\epsilon$ close to the solution of the corresponding ODE. However, in order to achieve the same (up to a log factor) sample complexity as in batch TD learning, the communication complexity may also be significantly worse than that of batch TD learning in theory. But experimental results show a comparable performance of local TD algorithm with batch algorithm in terms of communication rounds.

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

## A   APPENDIX

In the section, we provide the experiment setup and additional experimental results for the synthetic setting.

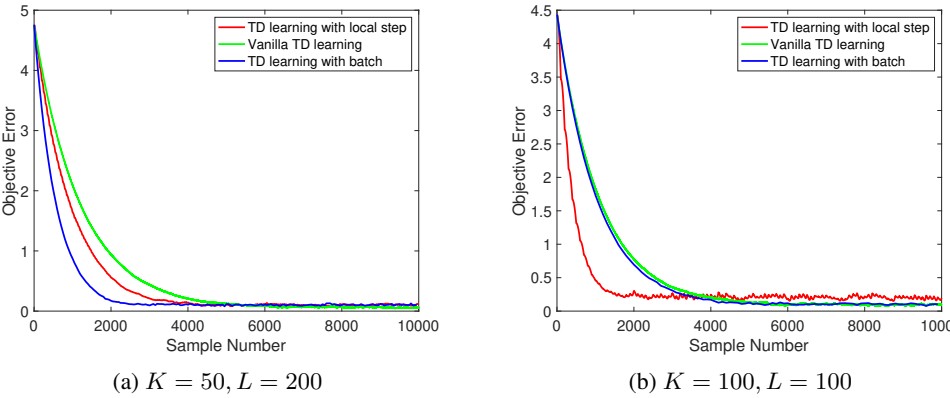

(a) $K = 50, L = 200$            (b) $K = 100, L = 100$

Figure 3: TD with local steps vs. vanilla TD and batched TD.

## A.1 Synthetic Experiment Setup

We consider the same setting as in Section 6.1 of (Zhang et al., 2018). There are $N = 20$ agents, each of which has a binary-valued action space, i.e., $\mathcal{A}^i = \{0, 1\}$ for all $i \in \mathcal{N}$. There are $|\mathcal{S}| = 10$ states. The entries in the transition matrix are uniformly sampled from the interval $[0, 1]$ and normalized to be stochastic. For each agent $i$ and global state action pair $(s, a)$, the mean reward $R^i(s, a)$ is sampled uniformly from $[0, 4]$ and the instantaneous rewards $\{r_t^i\}$ are sampled uniformly within the set $[R^i(s, a) - 0.5, R^i(s, a) + 0.5]$. The policy considered in the simulation is $\pi^i(\cdot|s) = 0.5$ for all $i \in \mathcal{N}$, $s \in \mathcal{S}$. The entries of feature matrix $\Phi$ are sampled uniformly at random from $[0, 1]$ with feature dimension $d = 5$ and ensured to be full rank. In addition, we set each feature vector to be unit length. The discount factor $\gamma$ is set to be 0.99. The network topology is chosen as a ring network with diagonal elements being 0.4 and off-diagonal elements being 0.3. The simulation results are averaged over 10 trials. We choose the step sizes for our algorithm to be 0.005, vanilla TD to be 0.1, and batched TD to be 10. We note that these step sizes are chosen to be best for the corresponding algorithms.

The objective error is defined as the normalized version of convergence term (LHS of the equation 9), i.e., the empirical sample mean errors divided by the number of agents $N$ and the dimension number $d$:

$$\text{Objective Error} := \text{sample average of } \frac{\sqrt{\sum_{i=1}^N \|w_{l,k}^i - w^*\|^2}}{dN} \text{ for 10 trials.}$$

We remark that due to the fact that the transition matrix is not dependent on joint action, the steady state distribution can be computed and so is the value of $w^*$, whose definition is in equation 4.

## A.2 Convergence Performance

First, we show that the local TD-update algorithm exhibits similar empirical convergence performances as the other algorithms.

In Fig. 3, we choose the number of local steps to be $K = 50$ and communication rounds to be $L = 200$ for the local TD-update algorithm. We choose batch size to be 50 and communication rounds to be 200 for batched TD algorithm in Fig. 3(a). In Fig. 3(b), we increase the number of local steps to $K = 100$ and decrease the communication rounds to $L = 100$ for the local TD-update algorithm, and we also increase batch size to be 100 and decrease the communication rounds to be 100 for batched TD algorithm. Note that in these parameter settings, our algorithm has the same number of communication rounds as the batched TD algorithm.

From both Figure 3(a) and (b), we can see that all algorithms converge to similar low objective error levels. This verifies the theoretical analysis that allowing local steps and performing infrequent communications is feasible and can converge. Then, we look at the number of samples needed to reach the low objective error. From Figure 3(a), we see that when local steps are relatively small, i.e., $K = 50$, the vanilla TD algorithm requires the most samples, while batched TD requires the

least, and our local TD algorithm with local steps requires somewhere in between. In Figure 3(b), when we increase the local steps to be $K = 100$, the number of samples of the local TD-update algorithm requires is the least compared to the other two algorithms. On the other hand, when the batch size increases, the number of samples required for batched TD to converge to low objective error is very similar to that of vanilla TD. However, we note that both our algorithm and batched TD requires less than 200 rounds of communication to reach that level of objective error in Figure 3(a) and less than 100 rounds of communication in Figure 3(b). Moreover, these results show that the local TD-update approach can effectively lower communication complexity of MARL-PE compared to the vanilla TD method.

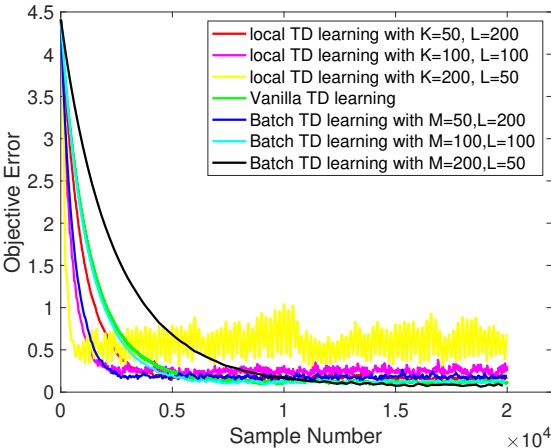

Figure 4: Additional TD with local steps vs. vanilla TD and batched TD.

In addition, we compare different pairs of local step $K$ and communication round $L$ of our proposed algorithm with different pairs of batch size $M$ and communication round $L$ in Figure 4. In general, local TD learning performs at a comparable level with batch TD algorithm, especially for the case $K = M = 50$ and $L = 200$, red curve for local TD algorithm and blue curve for the batch TD algorithm respectively. When increasing the local steps for local TD algorithm, it converges faster but to a higher error level. In comparison, when increasing the batch size for batch TD algorithm, the convergence is slightly slower but to a smaller error level.

### A.3   IMPACTS OF THE NUMBER OF LOCAL STEPS ON CONVERGENCE

Next, we illustrate the effect of the number of local steps on the convergence for our proposed algorithm. In Figure 5, we vary the number of local steps from $K = 40$ to $K = 250$. There are two interesting observations from this result. First, the number of samples needed to reach a low level objective error decreases as the number of local steps increases. For example, when $K = 100$ or larger, the curves drop much more rapidly in the beginning compared to the curves with smaller local steps. Second, the level of objective error that the algorithm converges to increases as the number of local steps increases. For example, when $K \leq 100$, the objective error is relatively low and stable, but as $K$ increases to 200 or 250, the objective error is large and oscillates around a relatively higher level. This observation is consistent with our theoretical analysis. Recall the second term on the RHS of equation 7 in Lemma 1 that is proportional to the product of step size $\beta$ and local step $K$. This term says the objective error will converge to neighborhood of zero, for which the size of the neighborhood depends on $\beta K$. As a result, for a larger $K$-value the objective error will oscillate with a larger magnitude. This resembles the constant step size term in the convergence of the SGD method in the distributed optimization problem (Nedic & Ozdaglar, 2009). Also intuitively, the agent-drift phenomenon will exacerbate with larger local steps, so the results for $K \geq 200$ in Figure 5 agree with such intuitions. In summary, empirically, when the step size is fixed, larger local steps helps the convergence speed, but it will also result in converging to a higher objective error level.

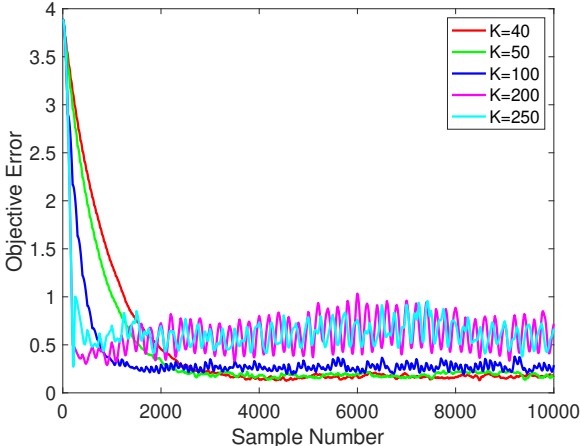

Figure 5: Impact of local steps $K$ on convergence.

### A.4 LINEAR CONVERGENCE WITH RESPECT TO SAMPLE NUMBER

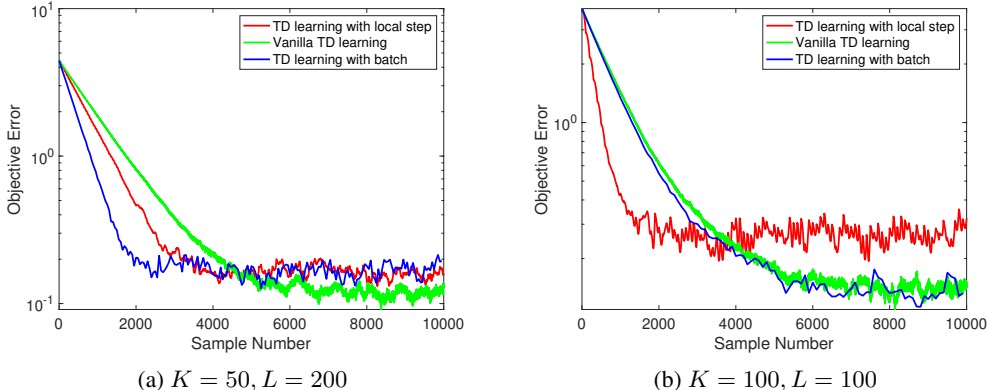

(a) $K = 50, L = 200$           (b) $K = 100, L = 100$

Figure 6: Convergence in Log Scales

In Figure 6, we provide the log scaled objective error with respect to sample number for all three algorithms. All algorithms behave similarly. Specifically, there are two phases in the convergences, roughly a linear convergence phase at first and oscillation phase at a relatively low objective error level. At the linear convergence phase, the objective error quickly decreases as the number of sample increases and reaches a relatively low objective error phase; at the oscillation phase, the objective error oscillates. This oscillation is likely caused by both the randomness in sample and the consensus error. For the local TD algorithm, as we increase the local step $K$ from 50 to 100, the only difference is the oscillation phase, where when $K = 100$, the objective error is higher due to the effective step sizes $\beta K$ becomes larger, as indicated by the second term in Equation equation 7. This constant error term that is linear with respect to step size is common in the constant step size learning scheme.

## B COOPERATIVE NAVIGATION TASK

### B.1 EXPERIMENT SETUP AND PERFORMANCE METRICS

We consider a cooperative navigation task that is adapted from one of Multi-Agent Environments (Lowe et al., 2017). There are $N = 9$ agents in total, and the goal is to cover 9 landmarks collaboratively. Each agent can choose from the action space, $\mathcal{A}^i =\{$no action, move left, move right, move down, move up$\}$, in order to reach the goals based on the policy. The policy considered in the simulation is $\pi^i(\cdot|s) = 0.2$ for all actions and $i \in \mathcal{N}$, $s \in \mathcal{S}$, i.e. uniformly random policy. The local rewards are given by the distance between the agents and goal landmarks. However, if they collide

with each other, there would be a penalty given. The agents are trained to cover landmarks and reach the destination while avoid colliding with other agents, and the entire training process is fully decentralized. Figure 2a illustrates the task that agents are going to accomplish. As shown in the figure, the blue circles are the agents, and the black crosses are the landmarks. Agents should learn to cover goal landmarks, for example, following the arrows to finish the task. The feature dimension here is 36, i.e. $d = 36$, which includes all agents' self velocity, self position, landmark relative positions, other agent relative positions, and corresponding values are ranged within $(-\infty, \infty)$. Discount factor $\gamma$ is set to be 0.95. We choose step size $\beta$ for our algorithm to be 0.05, for vanilla TD algorithm to be 0.1, and for batch TD algorithm to be 0.1 as well. We note that these step sizes are chosen for the best performance for the corresponding algorithms.

The performance metrics are mean squared Bellman error (MSBE) and consensus error (CE). The empirical squared Bellman error is defined as

$$\text{SBE}\left(\{w_k^i\}_{i=1}^N, s_k, s_{k+1}\right) := \frac{1}{N}\sum_{i\in\mathcal{N}}\left(\phi(s_k)^T w_k^i - \bar{r}_k - \gamma\phi(s_{k+1})^T w_k^i\right)^2,$$

where $\bar{r}_k = \frac{1}{N}\sum_{i\in\mathcal{N}} r_k^i$. Then, mean squared Bellman error is computed by averaging all squared Bellman error in the history, which is defined as

$$\text{MSBE} := \frac{1}{k}\sum_{\kappa=1}^k \text{SBE}\left(\{w_\kappa^i\}_{i=1}^N, s_\kappa, s_{\kappa+1}\right).$$

The consensus error is defined as

$$\text{CE}\left(\{w_k^i\}_{i=1}^N\right) := \frac{1}{N}\sum_{i\in\mathcal{N}}\left\|w_k^i - \overline{w}_k\right\|^2.$$

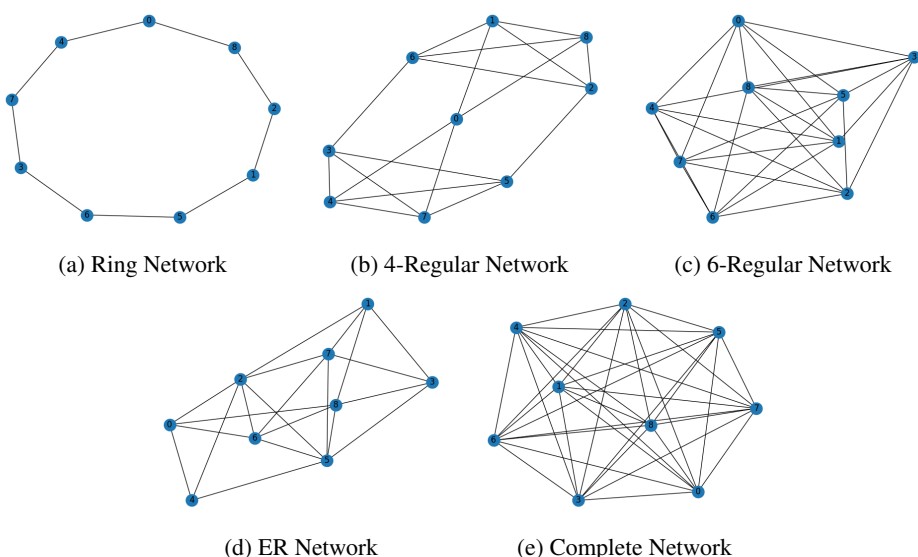

(a) Ring Network          (b) 4-Regular Network          (c) 6-Regular Network

(d) ER Network          (e) Complete Network

Figure 7: Network Topology

## B.2 Network Topology

We compare all algorithms under five different network typologies. These are ring network, 4-regular network, 6-regular network, Erdos-Renyi(ER) network with 0.5 connection probability, and fully connected network. These network topologies are illustrated in Figure 7. For simplicity, the local aggregation is the average of neighboring nodes for all networks.

## B.3 Convergence Performance

First, we show that the local TD-update algorithm exhibits similar empirical convergence performances as the other algorithms while using less communication rounds among agents. Here we

choose the number of local steps to be $K = 20$ for the local TD-update algorithm, the batch size to be 10 for the batched TD algorithm. For all of these algorithms, we set the total sample number to be 10000. The comparison among the algorithms is shown in Figure 9-13 over various network topologies.

We can see that all algorithms converge to very similar low squared Bellman error and consensus error level based on these figures. This verifies our analysis that allowing local steps and performing infrequent communications is feasible and can converge. In this case, vanilla TD algorithm performs 10000 communication rounds, which is the most, batch TD algorithm performs 1000 communication rounds, while local TD-update algorithm only performs 500 communication rounds. Left columns of the Figure 9-13 demonstrate the mean squared Bellman error, and right columns demonstrate the consensus error. For the mean squared Bellman error, the curve of local TD-update algorithm and the curve of vanilla TD algorithm has similar dropping rates, while the curve of batched TD algorithm drops slower. In terms of error level, all algorithms converge to a comparable mean squared Bellman error, which is around 2. On the other hand, the consensus errors in the right column of Figure 9-13, of all algorithms are close to 0, although local TD-update algorithm may have a relatively high error level depending on the network topology. In Figure 8, we present the topology effect on our proposed algorithm. It is, in general, as the network becomes more and more connected the consensus error fluctuates less. Intuitively, with better connected network, after local consensus aggregation, the parameter can be closer to the global average of the network.

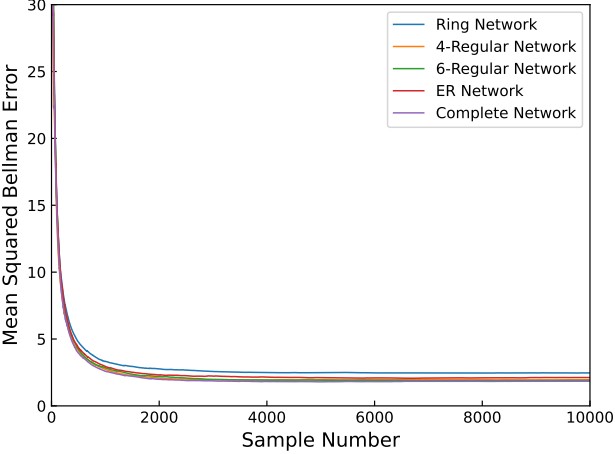

Figure 8: Topology on Local TD Algorithm

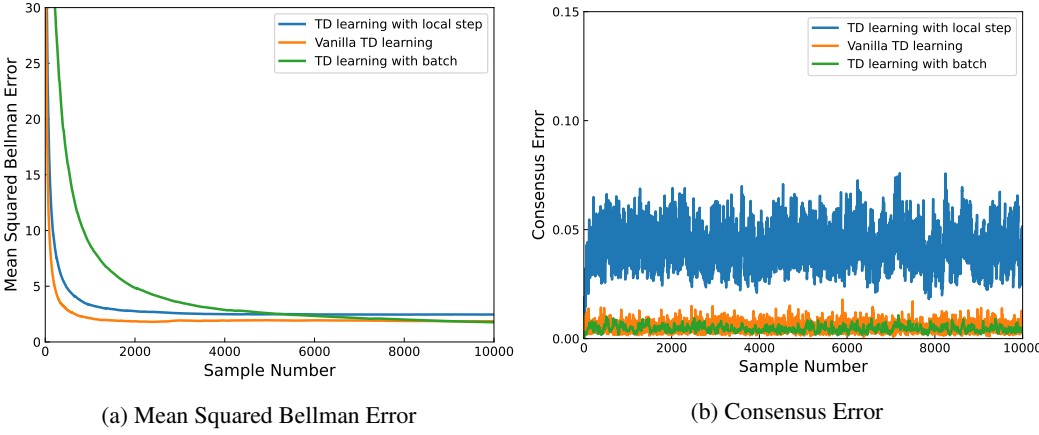

(a) Mean Squared Bellman Error

(b) Consensus Error

Figure 9: Comparison among Algorithms in Ring Network

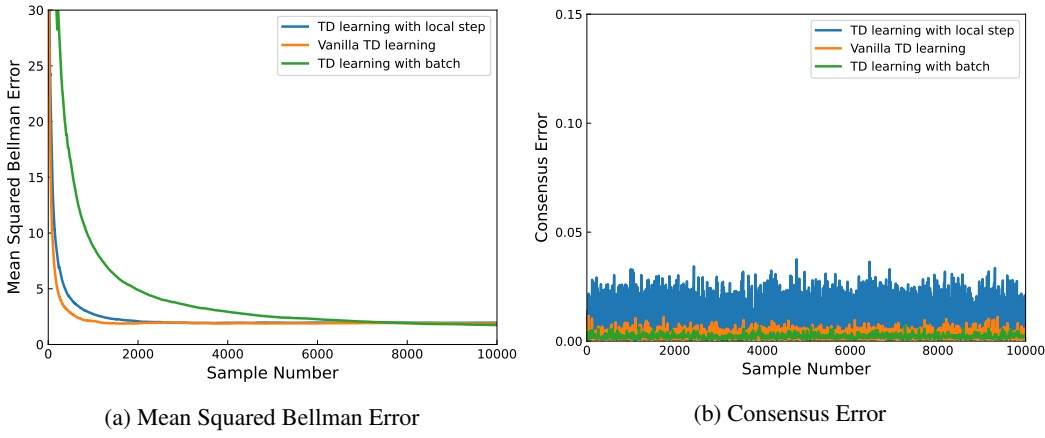

(a) Mean Squared Bellman Error

(b) Consensus Error

Figure 10: Comparison among Algorithms in 4-Regular Network

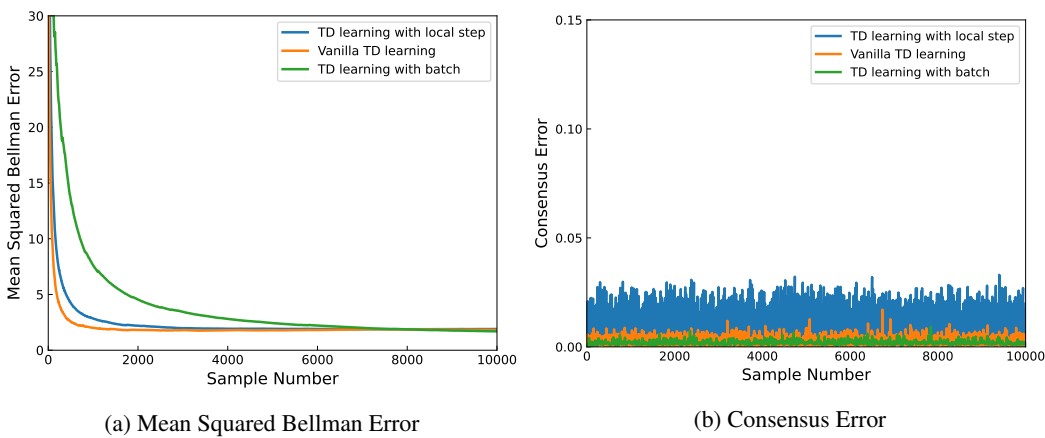

(a) Mean Squared Bellman Error

(b) Consensus Error

Figure 11: Comparison among Algorithms in 6-Regular Network

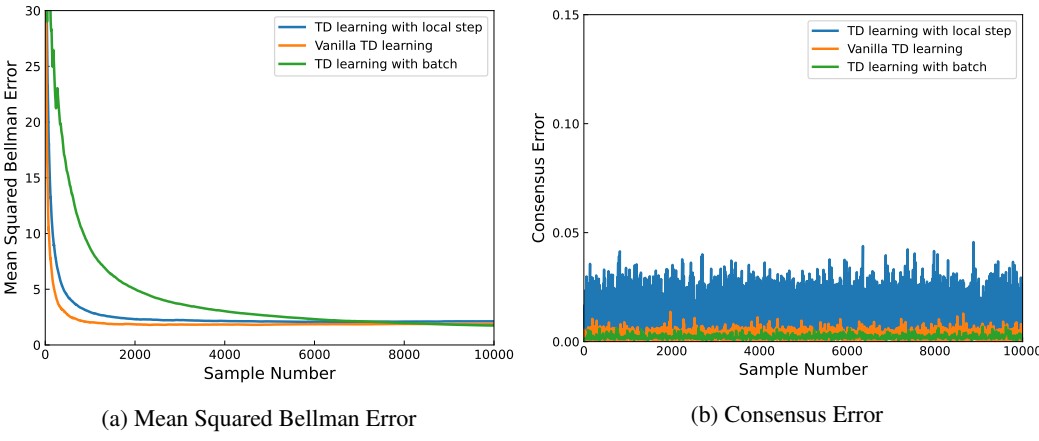

(a) Mean Squared Bellman Error

(b) Consensus Error

Figure 12: Comparison among Algorithms in ER Network

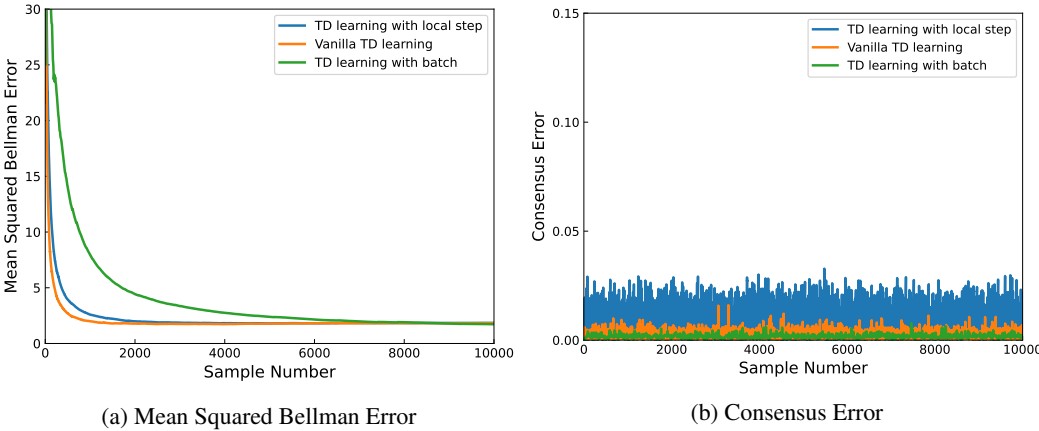

(a) Mean Squared Bellman Error

(b) Consensus Error

Figure 13: Comparison among Algorithms in Complete Network

In addition, we have compared different pairs of local step and communication round for our proposed algorithm and different pairs of batch size and communication round for batch TD algorithm in Figure 14 for cooperative navigation task. In this setting, all algorithms converge to a similar Bellman error level. However, the convergence for local TD algorithm is faster than batch TD algorithms in all parameter settings. Moreover, with the increase of batch size, batch TD algorithm seems to converge slower while with the increase of local step, local TD algorithm convergence increases in general but not significantly.

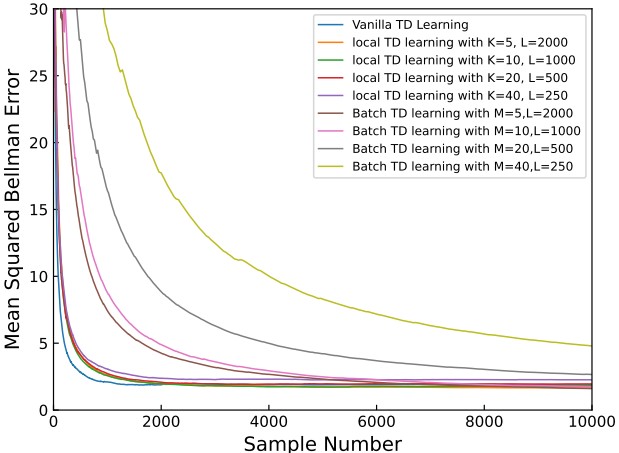

Figure 14: Additional Comparisons

## B.4    CONVERGENCE PERFORMANCE WITH RESPECT TO COMMUNICATION ROUNDS

In Figure 15, we provide the convergence results with respect to the communication rounds for all algorithms, where the local step $K = 20$ for local TD-update algorithm and the batch size is 10 for batch TD algorithm. We can see that within 500 communication rounds, local TD-update algorithm converges faster than batched TD algorithm and vanilla TD algorithm to a very similar error level. Within 1000 communication rounds, batched TD algorithm and vanilla TD algorithm performs similarly. The performance of these algorithms is not affected by the network topology being used.

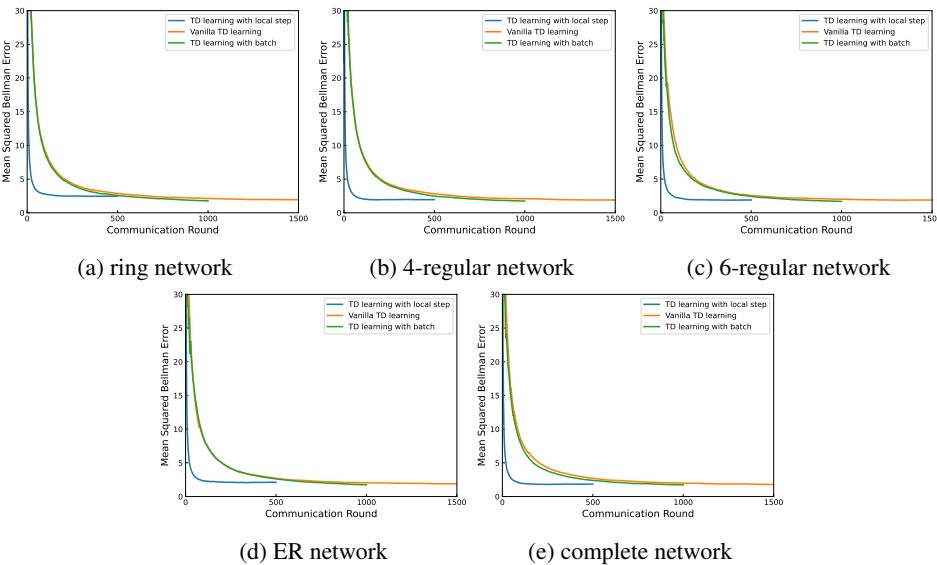

(a) ring network      (b) 4-regular network      (c) 6-regular network

(d) ER network      (e) complete network

Figure 15: Convergence Respect to Communication Rounds

### B.5 IMPACTS OF THE NUMBER OF LOCAL STEPS ON CONVERGENCE

Next, we illustrate the effect of the number of local steps $K$ on the convergence for our proposed algorithm. In Figure 16, we vary the number of local steps from $K = 10$ to $K = 200$. The right column is the consensus error of first 2000 samples, which displays a better view. As the number of local steps increases, the mean squared Bellman error converges to a higher error level, on the other hand, the consensus error oscillates more. In summary, larger local steps helps with saving more communication cost while it also result in converging to a higher mean squared Bellman error and a greater fluctuation of the consensus error. This conclusion also echoes with the synthetic experiment results in A.3.

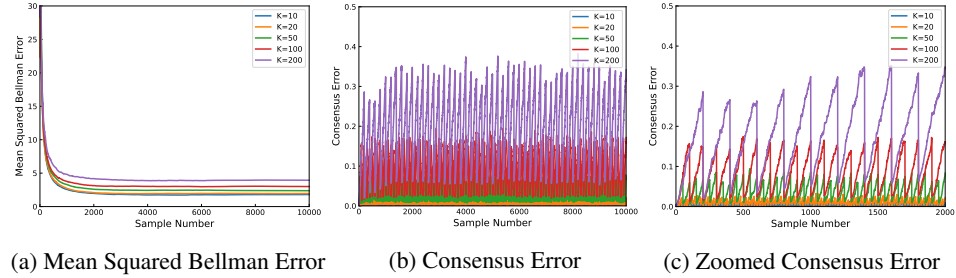

(a) Mean Squared Bellman Error      (b) Consensus Error      (c) Zoomed Consensus Error

Figure 16: Comparison among Different Local Steps in 4 Regular Network

### B.6 IMPACTS OF STEP SIZE ON CONVERGENCE

Here, we illustrate the effect of step size $\beta$ on the convergence for our proposed algorithm and batched TD algorithm over a 4-regular network. First of all, we can take a look at batch TD algorithm displayed on Figure 17-19 over various batch sizes. The brown line shows the performance of our proposed algorithm to help compare with batch TD algorithm. In general, in terms of mean squared Bellman error, larger step size $\beta$ leads to faster convergence speed and larger error level. Smaller step size $\beta$ leads to slower convergence speed, but could eventually converge to a smaller error level. Also, larger step size $\beta$ results in a greater consensus error. Thus, we set step size $\beta = 0.1$ and batch size to be 10 for batch TD algorithm.

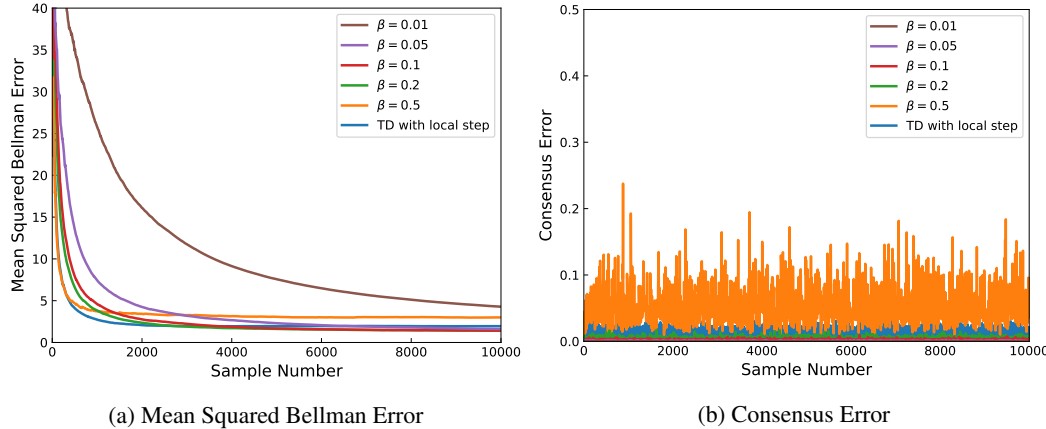

(a) Mean Squared Bellman Error

(b) Consensus Error

Figure 17: Comparison among Different step sizes $\beta$ for TD Learning with Batch Size 5 in 4 Regular Network

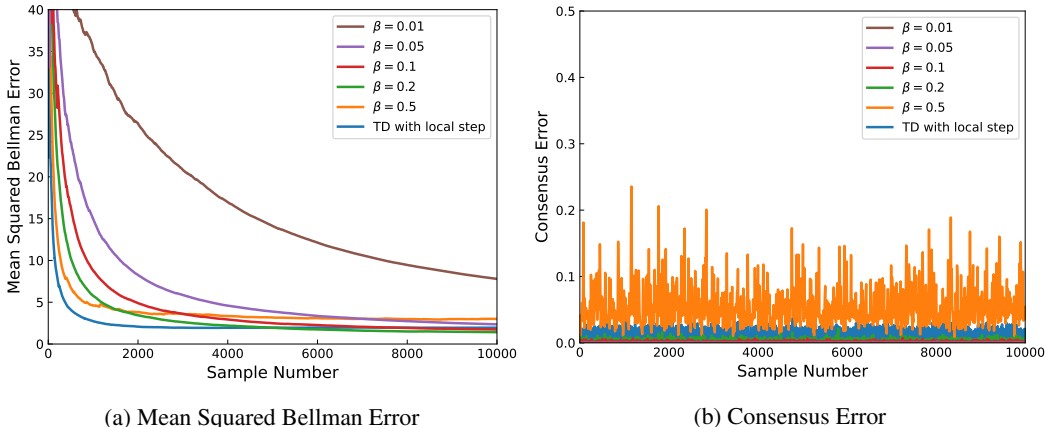

(a) Mean Squared Bellman Error

(b) Consensus Error

Figure 18: Comparison among Different step sizes $\beta$ for TD Learning with Batch Size 10 in 4 Regular Network

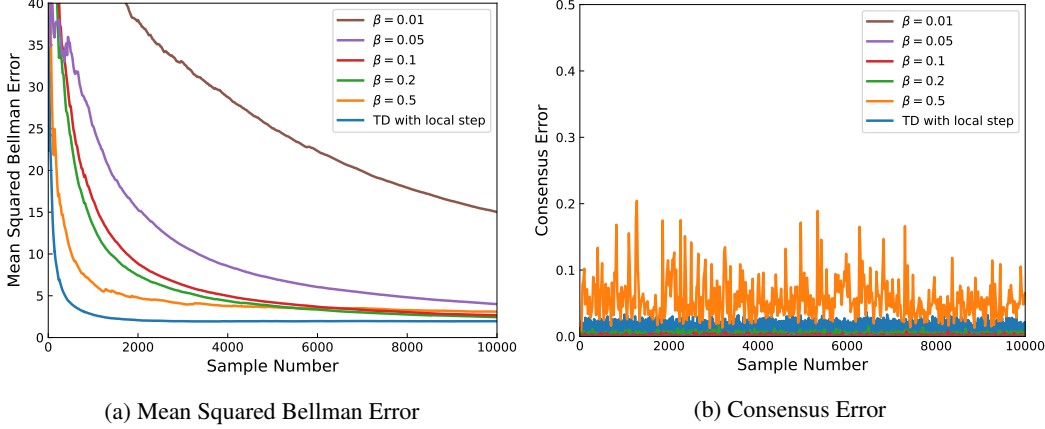

(a) Mean Squared Bellman Error

(b) Consensus Error

Figure 19: Comparison among Different step size $\beta$ for TD Learning with Batch Size 20 in 4 Regular Network

As shown in Figure 20-22, similar to batch TD algorithm, larger step size $\beta$ leads to faster convergence speed and larger mean squared Bellman and consensus error level in local TD-update algorithm. However, the error differences among different step sizes $\beta$ are not that significant compared

to batch TD algorithm. In order to balance among convergence speed, error level, and communication cost, we decide to use step size $\beta = 0.05$ and local step $K = 20$. In this way, we could have a fast convergence speed, low communication cost, and a relatively low error level.

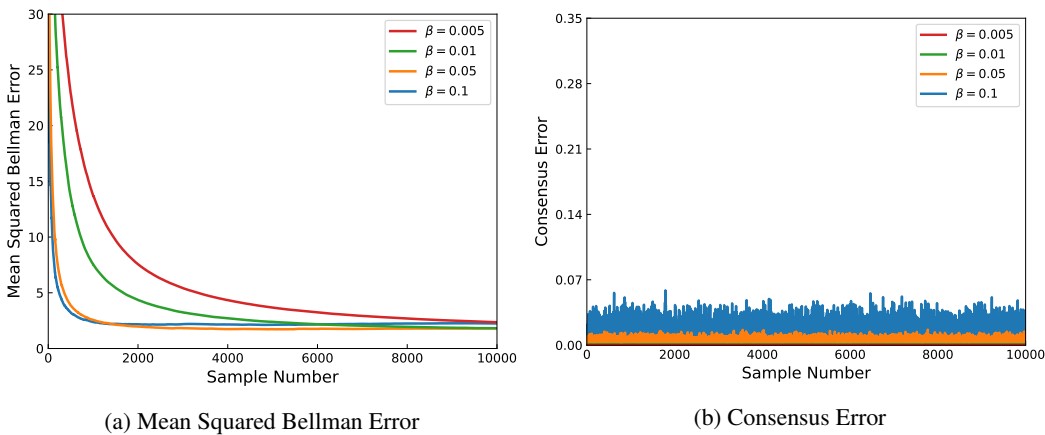

(a) Mean Squared Bellman Error

(b) Consensus Error

Figure 20: Comparison among Different step sizes $\beta$ for TD Learning with Local Step $K = 10$ in 4 Regular Network

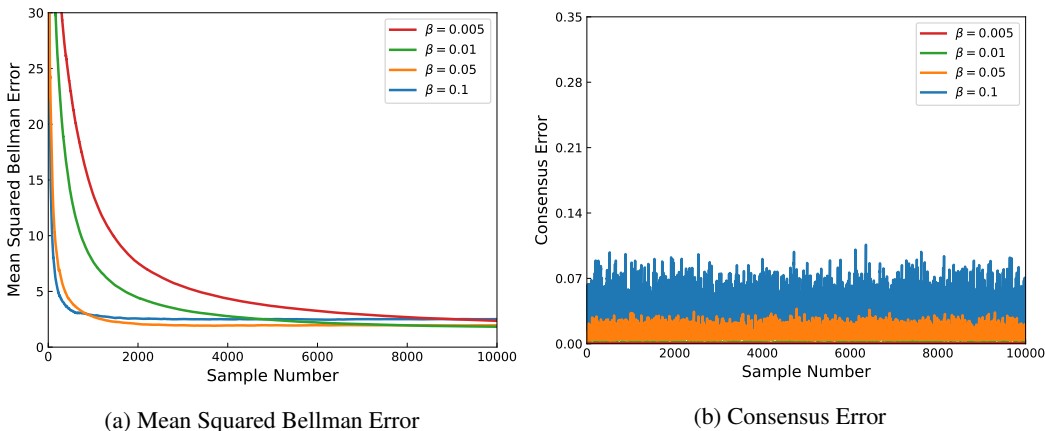

(a) Mean Squared Bellman Error

(b) Consensus Error

Figure 21: Comparison among Different step sizes $\beta$ for TD Learning with Local Step $K = 20$ in 4 Regular Network

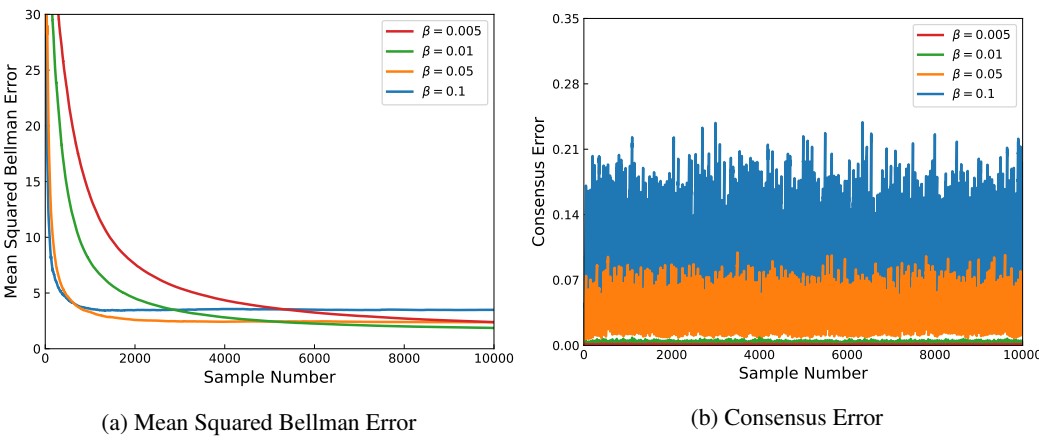

(a) Mean Squared Bellman Error

(b) Consensus Error

Figure 22: Comparison among Different step sizes $\beta$ for TD Learning with Local Step $K = 50$ in 4 Regular Network

# C PROOFS OF LEMMAS AND THEOREM

In this section, we provide the derivation for the consensus error. Then, we prove the Lemma 1, provide detailed discussion for Lemma 2 and the proof of Theorem 1.

## C.1 THE DERIVATION OF CONSENSUS ERROR

Within each communication round $0 \leq l \leq L - 1$, the parameter update $w_{l,k}^i$ for agent $i$ at local step $k$ can be written as follows

$$
\begin{aligned}
w_{l,k+1}^i &= w_{l,k}^i + \beta \delta_{l,k}^i \cdot \phi(s_{l,k}) \\
&= \left( I + \beta \phi(s_{l,k}) [\gamma \phi(s_{l,k+1}) - \phi(s_{l,k})]^T \right) w_{l,k}^i + \beta r_{l,k+1}^i \phi(s_{l,k}) \\
&= B_{l,k} w_{l,k}^i + c_{l,k}^i
\end{aligned}
\tag{10}
$$

where $B_{l,k} := I + \beta \phi(s_{l,k})[\gamma \phi(s_{l,k+1}) - \phi(s_{l,k})]^T$ and $c_{l,k}^i := \beta r_{l,k+1}^i \phi(s_{l,k})$. Then, from local step 0 to $K - 1$, we have

$$
w_{l,K}^i = \prod_{k=0}^{K-1} B_{l,k} w_{l,0}^i + \sum_{k=0}^{K-1} \prod_{n=k+1}^{K-1} B_{l,n} c_{l,k}^i.
$$

After a consensus update, the parameter for agent $i$ will be

$$
\begin{aligned}
w_{l+1,0}^i &= \sum_{j \in \mathcal{N}_i} A(i,j) \cdot w_{l,K}^j \\
&= \sum_{j \in \mathcal{N}_i} A(i,j) \cdot \left( \prod_{k=0}^{K-1} B_{l,k} w_{l,0}^j + \sum_{k=0}^{K-1} \prod_{n=k+1}^{K-1} B_{l,n} c_{l,k}^j \right) \\
&= \sum_{j \in \mathcal{N}_i} A(i,j) \cdot \prod_{k=0}^{K-1} B_{l,k} w_{l,0}^j + \sum_{j \in \mathcal{N}_i} A(i,j) \cdot \sum_{k=0}^{K-1} \prod_{n=k+1}^{K-1} B_{l,n} c_{l,k}^j.
\end{aligned}
$$

The equation above shows the parameter update between two consecutive communication rounds. Now we consider the average dynamics of the parameters across all agents. Recall $\bar{w}_{l,k} = \frac{1}{N} \sum_{i \in \mathcal{N}} w_{l,k}^i$, then within each communication round $0 \leq l \leq L - 1$, using equation 10 we have

$$
\begin{aligned}
\bar{w}_{l,k+1} &= \frac{1}{N} \sum_{i \in \mathcal{N}} w_{l,k+1}^i \\
&= \frac{1}{N} \sum_{i \in \mathcal{N}} (B_{l,k} w_{l,k}^i + c_{l,k}^i) \\
&= B_{l,k} \bar{w}_{l,k} + \frac{1}{N} \sum_{i \in \mathcal{N}} c_{l,k}^i \\
&= B_{l,k} \bar{w}_{l,k} + \bar{c}_{l,k}
\end{aligned}
\tag{11}
$$

where $\bar{c}_{l,k} := \frac{1}{N} \sum_{i \in \mathcal{N}} c_{l,k}^i$. Hence, the average dynamics from local step 0 to $K - 1$ will be

$$
\bar{w}_{l,K} = \prod_{k=0}^{K-1} B_{l,k} \bar{w}_{l,0} + \sum_{k=0}^{K-1} \prod_{n=k+1}^{K-1} B_{l,n} \bar{c}_{l,k}.
$$

After a consensus update, we have

$$
\bar{w}_{l+1,0} = \bar{w}_{l,K}.
\tag{12}
$$

Note that the equation above means that consensus step will not change the average dynamics and average dynamic will only be updated during local steps.

For an agent $i \in \mathcal{N}$, we consider the consensus error at communication round $l$ and local step $k$, where $0 \leq k \leq K - 1$ and recall $Q_{l,k}^i = w_{l,k}^i - \bar{w}_{l,k}$. Then, we have

$$\begin{aligned}
Q_{l,k+1}^i &= w_{l,k+1}^i - \bar{w}_{l,k+1} \\
&= B_{l,k} w_{l,k}^i + c_{l,k}^i - B_{l,k} \bar{w}_{l,k} - \bar{c}_{l,k} \\
&= B_{l,k}(w_{l,k}^i - \bar{w}_{l,k}) + c_{l,k}^i - \bar{c}_{l,k} \\
&= B_{l,k} Q_{l,k}^i + c_{l,k}^i - \bar{c}_{l,k}.
\end{aligned}$$

Then, for the matrix form $Q_{l,k} = [Q_{l,k}^1, \cdots, Q_{l,k}^N] \in R^{d \times N}$, we have

$$Q_{l,k+1} = B_{l,k} Q_{l,k} + C_{l,k}(I - \frac{1}{N} \mathbf{1} \mathbf{1}^T)$$

where $C_{l,k} := [c_{l,k}^1 \cdots c_{l,k}^N]$ and $\mathbf{1}$ denotes the all-1 column vector. Then, for communication round $l$, we have

$$Q_{l,K} = \prod_{k=0}^{K-1} B_{l,k} Q_{l,0} + \sum_{t=0}^{K-1} \prod_{\tilde{t}>t}^{K-1} B_{l,\tilde{t}} C_{l,k}(I - \frac{1}{N} \mathbf{1} \mathbf{1}^T)$$

After a consensus update, we have

$$\begin{aligned}
w_{l+1,0}^i - \bar{w}_{l,k} &= \sum_{j \in \mathcal{N}_i} A(i,j) w_{l,K}^j - \bar{w}_{l,K} \\
&= \sum_{j \in \mathcal{N}_i} A(i,j)(w_{l,K}^j - \bar{w}_{l,K}) = \sum_{j \in \mathcal{N}_i} A(i,j) Q_{l,K}^j.
\end{aligned}$$

As a result, we have

$$\begin{aligned}
Q_{l+1,0} &= Q_{l,K} A^T \\
&= \prod_{k=0}^{K-1} B_{l,k} Q_{l,0} A^T + \sum_{t=0}^{K-1} \prod_{\tilde{t}>t}^{K-1} B_{l,\tilde{t}} C_{l,t}(I - \frac{1}{N} \mathbf{1} \mathbf{1}^T) A^T.
\end{aligned}$$

After $L$ communication rounds, we have

$$Q_{L,0} = \prod_{l=0}^{L-1} \prod_{k=0}^{K-1} B_{l,k} Q_{0,0}(A^T)^L + \sum_{l=0}^{L-1} \prod_{j=1}^{L-1-l} \prod_{k=0}^{K-1} B_{l+j,k} \sum_{t=0}^{K-1} \prod_{\tilde{t}>t}^{K-1} B_{l,\tilde{t}} C_{l,t}(I - \frac{1}{N} \mathbf{1} \mathbf{1}^T)(A^T)^{L-l}$$

Note that for the second term when $l = L - 1$, inside the summation, the summand becomes $\sum_{t=0}^{K-1} \prod_{\tilde{t}>t}^{K-1} B_{L-1,\tilde{t}} C_{L-1,t}(I - \frac{1}{N} \mathbf{1} \mathbf{1}^T) A^T$. In other words, the matrix multiplier in front becomes an identity matrix.

## C.2 PROOF OF LEMMA 1

The norm of the consensus error is following

$$\begin{aligned}
&\|Q_{L,0}\| \\
=&\| \prod_{l=0}^{L-1} \prod_{k=0}^{K-1} B_{l,k} Q_{0,0}(A^T)^L + \sum_{l=0}^{L-1} \prod_{j=1}^{L-1-l} \prod_{k=0}^{K-1} B_{l+j,k} \sum_{t=0}^{K-1} \prod_{\tilde{t}>t}^{K-1} B_{l,\tilde{t}} C_{l,t}(I - \frac{1}{N} \mathbf{1} \mathbf{1}^T)(A^T)^{L-l}\| \\
\leq&\| \prod_{l=0}^{L-1} \prod_{k=0}^{K-1} B_{l,k} Q_{0,0}(A^T)^L\| \\
&+ \| \sum_{l=0}^{L-1} \prod_{j=1}^{L-1-l} \prod_{k=0}^{K-1} B_{l+j,k} \sum_{t=0}^{K-1} \prod_{\tilde{t}>t}^{K-1} B_{l,\tilde{t}} C_{l,t}(I - \frac{1}{N} \mathbf{1} \mathbf{1}^T)(A^T)^{L-l}\|.
\end{aligned} \tag{13}$$

Before obtaining bounds on the terms of the consensus error in equation 13, we first provide some useful bounds on $B_{l,k}$ and $C_{l,k}$. First, we have

$$
\begin{aligned}
||B_{l,k}|| &= ||I + \beta\phi(s_{l,k})[\gamma\phi(s_{l,k+1}) - \phi(s_{l,k})]^T|| \\
&\leq 1 + \beta||\phi(s_{l,k})||(\gamma||\phi(s_{l,k+1})|| + ||\phi(s_{l,k})]^T||) \\
&\leq 1 + \beta(1+\gamma) < 1 + 2\beta
\end{aligned}
$$

where the second inequality is due to Assumption 4. Then, we have $||C_{k,l}|| \leq \beta\sqrt{N}r_{\max}$, where $r_{\max} = \sup_{i,s,a} r^i(s,a)$ by Assumption 2. This is because

$$
\begin{aligned}
||C_{k,l}|| &= ||\beta\phi(s_{k,l})(r^1_{k,l+1}, \cdots, r^N_{l,k+1})|| \\
&\leq \beta||\phi(s_{k,l})|| \cdot ||(r^1_{k,l+1}, \cdots, r^N_{l,k+1})|| \\
&= \beta\sqrt{N}r_{\max}.
\end{aligned}
$$

Next, inspired by Srikant & Ying (2019), we want to use the following bound

$$
(1+x)^K \leq 1 + 2xK
$$

for small $x$. Note that

$$
(1+x)^K|_{x=0} = 1 + 2xK|_{x=0}
$$

and when $x \leq \frac{\log 2}{K-1}$,

$$
\frac{\partial}{\partial x}(1+x)^K = K(1+x)^{K-1} \leq Ke^{x(K-1)} \leq 2K = \frac{\partial}{\partial x}(1+2xK)
$$

where the first inequality is due to the fact $\log(1+x) \leq x$ for $x \geq 0$ and the second inequality is due to the fact $x \leq \frac{\log 2}{K-1}$. Let $2\beta = x$ and $\beta \leq \frac{1}{2K} \leq \frac{\log 2}{2(K-1)}$.

For the first term in equation 13, when $\beta \leq \frac{1}{2K}$, we have that

$$
\begin{aligned}
||\prod_{l=0}^{L-1}\prod_{k=0}^{K-1} B_{l,k}Q_{0,0}(A^T)^L|| &\leq ||\prod_{l=0}^{L-1}\prod_{k=0}^{K-1} B_{l,k}|| \cdot ||Q_{0,0}(A^T)^L|| \\
&\leq \kappa(1+2\beta)^{KL}(1-\eta^{N-1})^L \\
&\leq \kappa(1+4\beta K)^L(1-\eta^{N-1})^L \\
&= \kappa\rho^L
\end{aligned}
$$

where we define $\rho := (1+4\beta K)(1-\eta^{N-1})$. When $0 < \beta K < \min\{\frac{1}{2}, \frac{\eta^{N-1}}{4(1-\eta^{N-1})}\}$, we have $0 < \rho < 1$. The second inequality comes from the following two results.

First, consider the case where $A$ is a symmetric matrix for simplicity, then we have

$$
\begin{aligned}
||Q_{0,0}A^L_{1,:}|| &= ||Q_{0,0}A^L_{1,:} - Q_{0,0}\frac{1}{N}\mathbf{1}|| \\
&= ||\sum_{i\in\mathcal{N}}(A^L_{1,i} - \frac{1}{N})Q^i_{0,0}|| \\
&\leq \sum_{i\in\mathcal{N}}|A^L_{1,i} - \frac{1}{N}| \cdot ||Q^i_{0,0}|| \\
&\leq N \cdot 2\frac{1+\eta^{-(N-1)}}{1-\eta^{N-1}}(1-\eta^{N-1})^L \cdot \max_{i\in\mathcal{N}}||Q^i_{0,0}|| \\
&\leq 2N\frac{1+\eta^{-(N-1)}}{1-\eta^{N-1}}(1-\eta^{N-1})^L \cdot ||Q_{0,0}||,
\end{aligned}
$$

where the second inequality is from Nedic & Ozdaglar (2009) (Proposition 1). Hence, $||Q_{0,0}A^L|| \leq 2N^2\frac{1+\eta^{-(N-1)}}{1-\eta^{N-1}}(1-\eta^{N-1})^L \cdot ||Q_{0,0}|| = \kappa_1(1-\eta^{N-1})^L||Q_{0,0}||$, where $\kappa_1 = 2N^2\frac{1+\eta^{-(N-1)}}{1-\eta^{N-1}}$.

Second, we have

$$||\prod_{l=0}^{L-1}\prod_{k=0}^{K-1}B_{l,k}|| \le \prod_{l=0}^{L-1}\prod_{k=0}^{K-1}||B_{l,k}||$$
$$\le \prod_{l=0}^{L-1}\prod_{k=0}^{K-1}(1+2\beta) = (1+2\beta)^{KL}.$$

To bound the second term of equation 13, we have

$$||(I-\frac{1}{N}\mathbf{1}\mathbf{1}^T)A^{L-l}|| = ||A^{L-l} - \frac{1}{N}\mathbf{1}\mathbf{1}^T|| \le 2N^2(1+\eta^{-(N-1)})(1-\eta^{N-1})^{L-l-1}.$$

where the inequality is also from Nedic & Ozdaglar (2009) (Proposition 1). Then, we also have

$$\sum_{t=0}^{K-1}\prod_{\tilde{t}>t}^{K-1}||B_{l,\tilde{t}}|| \cdot ||C_{l,t}||$$
$$\le \sum_{t=0}^{K-1}(1+2\beta)^{K-1-t} \cdot \beta\sqrt{N}r_{\max}$$
$$=\beta\sqrt{N}r_{\max}\sum_{t=0}^{K-1}(1+2\beta)^{K-1-t}$$
$$\le 2\beta K\sqrt{N}r_{\max}.$$

Then, for the multipliers, we have

$$||\prod_{j=1}^{L-1-l}\prod_{k=0}^{K-1}B_{l+j,k}|| \le (1+2\beta)^{(L-l-1)K} \le (1+4\beta K)^{L-l-1}.$$

Finally, for the second term in consensus error equation 13, we have

$$||\sum_{l=0}^{L-1}\prod_{j=1}^{L-1-l}\prod_{k=0}^{K-1}B_{l+j,k}\sum_{t=0}^{K-1}\prod_{\tilde{t}>t}^{K-1}B_{l,\tilde{t}}C_{l,t}(I-\frac{1}{N}\mathbf{1}\mathbf{1}^T)(A^T)^{L-l}||$$
$$\le \sum_{l=0}^{L-1}||\prod_{j=1}^{L-1-l}\prod_{k=0}^{K-1}B_{l+j,k}|| \cdot ||\sum_{t=0}^{K-1}\prod_{\tilde{t}>t}^{K-1}B_{l,\tilde{t}}C_{l,t}|| \cdot ||(I-\frac{1}{N}\mathbf{1}\mathbf{1}^T)(A^T)^{L-l}||$$
$$\le \sum_{l=0}^{L-1}(1+4\beta K)^{L-l-1} \cdot 2\beta K\sqrt{N}r_{\max} \cdot 2N^2(1+\eta^{-(N-1)})(1-\eta^{N-1})^{L-l-1}$$
$$\le \kappa_2\beta K\sum_{l=0}^{L-1}\rho^{L-l-1}$$
$$\le \frac{\kappa_2\beta K}{1-\rho}$$

where $\kappa_2 = 4(1+\eta^{-(N-1)})N^{\frac{5}{2}}r_{\max}$.

As a result, we have the results consensus bound of equation 7 in Lemma 1.

## C.3 DETAILS ON LEMMA 2

Recall the average parameter dynamic from equation 11, which is in among consecutive communication rounds, and equation 12, after a consensus update. The average parameter is signaled by the true average of the rewards, i.e. $\bar{r}_{l,k} = \frac{1}{N}\sum_{i\in\mathcal{N}}r_{l,k}^i$ during both the local TD steps and consensus steps. Therefore, it behaves like conventional TD update in a single agent setting. By Theorem 7 in Srikant & Ying (2019), we have following results

$$\mathbb{E}[||\bar{w}_{L,0}-w^*||^2] \le c_2(1-c_1\beta)^{KL-\tau}(||\bar{w}_0-w^*||+\frac{r_{\max}}{3})^2 + c_3\beta\tau.$$

where $\tau$ is a mixing time. Under Assumption 1, $\tau = O(\log \frac{1}{\beta})$. To specify the constants $c_1, c_2, c_3$, we introduce the Lyapunov equation. Recall the definition of $\Psi$ in equation 3, which is negative definite Tsitsiklis & Van Roy (1997). So we have a symmetric matrix $U > 0$ Srikant & Ying (2019) such that

$$\Psi^T U + U\Psi + I = 0$$

which is called the Lyapunov equation. For symmetric matrix $U$, there exist largest and smallest eigenvalues $\lambda_{\max}$ and $\lambda_{\min}$ respectively. In addition, $\lambda_{\max}$ and $\lambda_{\min}$ are both positive. So, the constants are

$$c_1 = \frac{0.9}{\lambda_{\max}},$$
$$c_2 = 2.25 \frac{\lambda_{\max}}{\lambda_{\min}},$$
$$c_3 = \frac{2\lambda_{\max}^2 (r_{\max}^2 + 55(1 + r_{\max})^3)}{0.9\lambda_{\min}}.$$

### C.4    PROOF OF THE THEOREM 1

For the mean square error, we have

$$\mathbb{E}[\sum_{i=1}^{N} \|w_{L,0}^i - w^*\|^2]$$
$$= \mathbb{E}[\sum_{i=1}^{N} \|w_{L,0}^i - \bar{w}_{L,0} + \bar{w}_{L,0} - w^*\|^2]$$
$$\leq 2\mathbb{E}[\sum_{i=1}^{N} \|w_{L,0}^i - \bar{w}_{L,0}\|^2] + 2\mathbb{E}[\sum_{i=1}^{N} \|\bar{w}_{L,0} - w^*\|^2]$$
$$\leq 2d\mathbb{E}[\|Q_{L,0}\|^2] + 2N\mathbb{E}[\|\bar{w}_{L,0} - w^*\|^2] \tag{14}$$

where the first inequality is due to $\|x + y\|^2 \leq 2\|x\|^2 + 2\|y\|^2$ and the second inequality $\|X\|_F \leq \sqrt{d}\|X\|$ for $X \in \mathbb{R}^{d \times N}$. Then, the stated result in equation 9 follows from Lemmas 1 and 2, and equation 14.

This concludes the proof.

