# OpenReview forum: "Achieving Communication-Efficient Policy Evaluation for Multi-Agent Reinforcement Learning: Local TD-Steps or Batching?"
_ICLR.cc/2023/Conference — Submitted to ICLR 2023_

### Official Review · Reviewer_stTJ · 2022-10-25

**Confidence:** 4
**Correctness:** 3
**Technical Novelty And Significance:** 2
**Empirical Novelty And Significance:** 3
**Recommendation:** 5

**Clarity, Quality, Novelty And Reproducibility:**

Clarity
It is difficult to hold the math or the experiments as the pillaring reference of clarity as the conditions under which the computed complexities hold empirically are not studied. Also, the K-rounds advantage of the local TD updates needs some discussion as to its performance gain over vanilla TD learning.
As for the writing, there is but one minor fragment, found on the last of the three bullet points on page 3: "Although the communication," seems cut off.

Quality
Including more varied experiments (perhaps, with a few of them violating some of the four assumptions) can be a strong upgrade to the current version.

Novelty
To the best of my knowledge, the presented work is original.

**Strength And Weaknesses:**

The paper addresses a well-defined niche. By allowing for multiple local TD updates, a more general version of the MARL-PE problem is constructed and its complexities calculated.

The argument made can be even stronger with i) a more comprehensive literature review and a more strategic positioning, ii) some attempts to provide insight as to the conditions under which local TD updates would perform at its computed worst complexity, and iii) discussion on the fairness of the tested algorithms in Figures 1 and 3, perhaps by including Figures 4 and 6 in the main paper.

It would be interesting to cover an important line of MARL research related to the exchange of state information, and especially the succinct packetization of state information to exchange. Works investigating emergent communication may prove important (and arguably necessary) contenders to the local TD updates in the sense that the message-generating function can also be considered a learnable parametrization. Whereas in this paper, that function would be just the identity function accepting the parameters (after local TD updates), studies such as SchedNet (ICLR 2019) would subject the individual state information to undergo feature extraction via a trainable logic (i.e., neural networks). With the common goal of improving communication efficiency, how does the presented method compare against emergent communication methods?

Investigation on the following questions would make the paper more insightful: Under what circumstances does the computed (worst-case) communication complexity actually manifest in experiments? Do those circumstances also affect whether batch learning performs at its computed worst?

Figures 1 and 3 can be better explained, with particular emphasis on the fairness of the tested algorithms. Since the x-axis is the communication rounds, Figures 1 and 3 do not correctly reflect the K > 1 "advantage" given to the local TD update method. At every x-axis tick, the proposed method runs for K (50 or 100 or etc.) iterations, so the graphical display can be misleading, unless accompanied by graphs such as Figures 4 and 6, which picture a more honest view by taking into account both K and L and plotting the errors against sample numbers. Indeed, it is sometimes the case (Fig. 4(a)) that batch learning does better than local TD updates, depending on the choice of K and L. If that really is the case, a further empirical analysis on the effects of K and L on the competition between batch learning and local TD updates may be an interesting addition.

**Summary Of The Paper:**

Authors present a generalized TD learning for the MARL setting, with local updates occurring for K > 1 iterations for each instance of communication. Sample complexities at the inner loop (local TD updates) and the outer loop (consensus updates) are computed based on mild assumptions. Theoretical formulation predicts a worse complexity for the outer loop for the TD update method to match that of the batch learning method, but experiments produce more optimistic numbers, for reasons not provided/studied in the paper.

**Summary Of The Review:**

The paper's main strength lies more in the analyses on the feasibility and comparability of local TD updates in MARL-PE than in designing and implementing a novel algorithm itself. If the paper is indeed best viewed in this light, then it would be pleasant to see some updates on the literature review, complexity analyses in the experiments, and the fairness in comparison.

---

> ### Author Response · Authors · 2022-11-19
> **Response to Reviewer stTJ**
>
> We sincerely thank the reviewer's constructive comments and valuable insights, which help improve the quality of our work significantly. We have carefully revised our paper according to your comments and suggestions. Please see our revised submission, where we have highlighted all major changes in **Red color**. Please also see our point-to-point responses as follows:
>
>
>
> > **Your Comment:** 1. The argument made can be even stronger with i) a more comprehensive literature review and a more strategic positioning,
>
>
>
> **Our Response:** Thank you for your suggestions. We have added more literature review in the related work section and positioned our paper more clearly.
>
>
>
> > **Your Commment:** 2. discussion on the fairness of the tested algorithms in Figures 1 and 3, perhaps by including Figures 4 and 6 in the main paper.
>
>
> **Our Response:** Thank you for your suggestions. Due to page limitation and for presentation clarity of the figures, we have to put one set of figures in the main paper, and the others in the appendix. In our original submission, we have put Figures 1 and 3 together in the main paper to emphasize the number of communication rounds required to converge for different algorithms. Following your suggestion, we have added the communication round figures and sample number figures together in Figure 1 of the main paper in this revision. We have also discussed the fair comparisons among algorithms in this revision.
>
> > **Your Commment:** 3. It would be interesting to cover an important line of MARL research related to the exchange of state information, and especially the succinct packetization of state information to exchange. Works investigating emergent communication may prove important (and arguably necessary) contenders to the local TD updates in the sense that the message-generating function can also be considered a learnable parametrization. Whereas in this paper, that function would be just the identity function accepting the parameters (after local TD updates), studies such as SchedNet (ICLR 2019) would subject the individual state information to undergo feature extraction via a trainable logic (i.e., neural networks). With the common goal of improving communication efficiency, how does the presented method compare against emergent communication methods?
>
> **Our Response:** Thanks for your comments and your pointer to a related work. SchedNet (ICLR 2019) and our work differs in the following key aspects:
>
> First of all, we want to emphasize the setting differences: in SchedNet (ICLR 2019), the paper adopted a so-called centralized-learning decentralized-execution scheme. In contrast, we considered a fully decentralized setting.
>
> Second,  we assume global states are observable to all agents, while SchedNet (ICLR 2019) considered a more general setting where only partial observations of the global states are available to each agent. Hence, communication is necessary in their setting.
>
> Third, about a common goal, SchedNet (ICLR 2019) considered constraints on each pair-wise communication channel to avoid collisions, which is a bottom-level communication protocol. This is different from the communication cost that we consider, which is the number of rounds of consensus operation among agents.
>
> We have added the above discussions in the related work section in this revision.
>
> > **Your Commment:** 4. Investigation on the following questions would make the paper more insightful: Under what circumstances does the computed (worst-case) communication complexity actually manifest in experiments? Do those circumstances also affect whether batch learning performs at its computed worst?
>
> **Our Response:** Thanks for your great question. At each iteration, our analysis for the convergence of the sum of all parameters to the fixed point of the ODE solution (the left-hand-side (LHS) of Eq. (9)) uses triangle inequality, where the two terms are the summation of distance from each parameter to the average parameter, i.e. the consensus error on LHS of Eq. (7) and the distance from average parameter to the ODE solution. The equality holds when mathematically the $w^{i}_t-\bar{w}_t$ equals to $\bar{w}_t-w^{*}$ for all $i\in\mathcal{N}$. However, due to the heterogeneous nature of our problem, this hardly holds true for all $t$ and with multiple local steps. As a result, the worst-case sample complexity in the Theorem is rare to manifest in practice. The same strategy of proof is used for the batch TD algorithm. As a result, the same circumstance will also apply in the batch TD algorithm that the upper bound is hard to manifest in practice.
>
> **Please continue to see our next response below.**

---

> > ### Author Response · Authors · 2022-11-19
> > **Response to Reviewer stTJ (Continued)**
> >
> >
> > > **Your Commment:** 5. Indeed, it is sometimes the case (Fig. 4(a)) that batch learning does better than local TD updates, depending on the choice of K and L. If that really is the case, a further empirical analysis on the effects of K and L on the competition between batch learning and local TD updates may be an interesting addition.
> >
> > **Our Response:** Thanks for your comment. We have added experiments and discussion on the additional choice of $K$ and $L$ for our proposed algorithm and batch size $M$ and communication around $L$ for batch TD algorithm in Figure 4 for the synthetic experiment and Figure 14 for the cooperative navigation task.
> >
> > > **Your Commment:** 6. As for the writing, there is but one minor fragment, found on the last of the three bullet points on page 3: "Although the communication," seems cut off.
> >
> > **Our Response:** Thanks for pointing it out. We have fixed it.

---

### Official Review · Reviewer_N76j · 2022-10-30

**Confidence:** 3
**Correctness:** 3
**Technical Novelty And Significance:** 2
**Empirical Novelty And Significance:** 2
**Recommendation:** 5

**Clarity, Quality, Novelty And Reproducibility:**

Overall the paper is written well. However, there are some ambiguity in technical challenges and main results. Although several technical challenges are mentioned, some of them have been addressed for naive TD methods in the MARL policy evaluation setting. So, it is not  very clear what new techniques the authors have extended. Although the agent-drift seems to be unique challenge, the authors haven't highlight how this fails naive application of recent TD analysis. For the algorithm analysis, it is important to clarify what is the new development for analyzing the proposed algorithm compared with the mean-path result from Srikant & Ying, 2019. More discussions on generalizations to other settings, e.g., nonlinear function approximation and other TD methods, would increase the impact of the study.

The paper has studied an extension of naive TD method for the MARL policy evaluation problem. Analyzing this extended method seems to be an adaption of existing TD analysis. Hence, novelty in method and analysis requires further justification.

Here are some other questions for consideration:

- 'agent drift' phenomenon is interesting. Is there a way to quantify it? It is important to characterize conditions when the proposed method works and fails.

- What is the specific condition for the inverse in Eq. (4) to exist?

- How does the mixing time in Eq. (5) relate to Assumption 1?




**Strength And Weaknesses:**

Strengths

- The communication compleixity is a bottleneck for distributed RL algorithms to efficiently work for large-scale networked systems. The studied multi-agent policy evaluation problem is a basic multi-agent RL task, which is real and meaningful.

- The authors particularly focus on a class of temporal learning (TD) methods with local TD updates per communication. Compared with batch-based TD methods, the studied method works simpler and is more useful in practice, since a large amount of batch sampling is expensive in many real-world applications.

- It is useful to establish near-optimal sample complexity of the proposed method as the well-studied batch-based TD methods. The studied communication complexity is also important to understanding pros/cons of the proposed methods.

 Weaknesses

- The proposed method is limited to an extension of naive TD methods for multi-agent MDPs. It is unknown about the generality of this method to other TD methods.

- This study is limited to the naive TD method which might not be very useful in some settings, for instance off-policy. Other limitations include the linear function approximation and the synchronous communication network.

- The study assumes a series of technical assumptions on the multi-agent MDPs and the function approximation. It is known that these assumptions provide convenience to analyze TD methods with Markovian data if the underlying MDP has a fast mixing. Hence, the first two claimed challenges are not the main difficulty of analyzing the proposed method.

- For the communication complexity, it is unknown if the proposed method can match batch-based TD methods. It is useful if some lower bound can be established for the proposed methods.

- It is expected to see that the information of network topology should be a big factor in the final error bound. This is missing. Experiments on other network topologies should be included for illustration.

- The literature review misses many recent works on MARL policy evaluation problems, which is detrimental to the novelty. A more complete comparison is recommended, e.g., the paper:  Taming Communication and Sample Complexities in Decentralized Policy Evaluation for Cooperative Multi-Agent Reinforcement Learning.


**Summary Of The Paper:**

The paper studies the multi-agent policy evaluation problem for a set of networked agents that can communicate through the network. In the linear function approximation regime, the authors apply the standard temporal difference (TD) learning method with no batch samples but multiple local TD updates per communication, and show near-optimal sample complexity and improved communication complexity compared to naive TD updates. Finally, the authors demonstrate the effectiveness of the proposed method in both synthetic and real multi-agent policy evaluation problems.

**Summary Of The Review:**

The paper studies an extension of naive TD method for the multi-agent policy evaluation problem. This extension is limited to a basic TD learning setting, which leaves a question about usefulness in other settings. The analysis also extends existing mean-path analysis in a more direct way, which leaves a question about novelty of analysis. Further study on conditions when the proposed method works and fails due to 'agent drift' is also needed. Since the authors miss many recent works, the novelty hasn't been fully established without a fair comparison. Generalizations to other settings haven't been established in either theory and experiments.

---

> ### Author Response · Authors · 2022-11-19
> **Response to Reviewer N76j**
>
> We sincerely thank the reviewer's constructive comments and valuable insights, which help improve the quality of our work significantly. We have carefully revised our paper according to your comments and suggestions. Please see our revised submission, where we have highlighted all major changes in **Red color**. Please also see our point-to-point responses as follows:
>
> > **Your Commment:** 1. The proposed method is limited to an extension of naive TD methods for multi-agent MDPs. It is unknown about the generality of this method to other TD methods.
>
> **Our Response:** Thanks for the great suggestion. The proposed method can be easily extended to TD($\lambda$) method. Specifically, algorithm-wise, we have added the following line after Line 7:
> $$ z_{l,k}=\gamma\lambda z_{l,k-1}+\phi(s_{l,k}) ,$$ which tracks the eligibility trace by taking a weighted average of the historic features. Thus, Line 9 will become
> $$w_{l,k+1}^{i}\leftarrow w_{l,k}^{i}+\beta \delta_{l,k}^{i}\cdot z_{l,k} .$$
> Note that when $\lambda=0$, it reduces to the TD(0) algorithm in our paper. Analysis-wise, the convergence can be established similarly to that of TD(0) in our paper. Other more sophisticated TD algorithms could also be explored as future work.
>
> > **Your Commment:** 2. This study is limited to the naive TD method which might not be very useful in some settings, for instance off-policy. Other limitations include the linear function approximation and the synchronous communication network.
>
> **Our Response:** Thank you for your insightful comment. First, we want to emphasize that our work is the first attempt towards investigating the feasibility of local steps in distributed TD learning and the performance comparison of the proposed method with vanilla TD and batch TD algorithms. That's why we start with considering the basic TD method.
>
> We note that, although beyond the current scope of this paper, it is possible to extend our "Local TD vs. Batching" analysis to the off-policy TD as you mentioned. Specifically, algorithm-wise, off-policy TD can be generalized into our local TD step setting by including important sampling. However, we note that when considering linear function approximation instead of the tabular method, off-policy training, bootstrapping (including our proposed method, vanilla TD and Batch TD) and function approximation form the notorious deadly triad [F]. As a result, the performance and convergence are largely unknown and require careful analysis, which is clearly beyond the scope of our current paper.
>
> [F] Sutton RS, Barto AG. Reinforcement learning: An introduction. MIT press; 2018 Nov 13.
>
> The reason we used linear function approximation is that we can provide a theoretical guarantee of convergence. We note that the tabular method is a special case of linear function approximation by setting the feature vector to be an appropriate unit vector. As for the nonlinear function approximation such as neural networks, to the best of our knowledge, the TD learning algorithm doesn't have a clear theoretical guarantee even in the single-agent setting.
>
> The synchronous communication network may not be ideal indeed. A future direction is that agent-side threshold triggered communication scheme, i.e. when an agent locally decides the change of parameter is significant enough, then it will send the parameters to the neighbors. However, this periodic synchronous communication network that we consider in our paper is possibly the simplest scheme of utilizing the local TD approach.
>
>
> **Please continue to see our next response below.**

---

> > ### Author Response · Authors · 2022-11-19
> > **Response to Reviewer N76j (Continued)**
> >
> > > **Your Commment:** 3. The study assumes a series of technical assumptions on the multi-agent MDPs and the function approximation. It is known that these assumptions provide convenience to analyze TD methods with Markovian data if the underlying MDP has a fast mixing. Hence, the first two claimed challenges are not the main difficulty of analyzing the proposed method.
> >
> > **Our Response:** Thanks for your comments. For the first two claimed challenges, we want to emphasize the difference between the TD-type of approach in MDP and the gradient-type of approach in optimization problems, rather than the difference between multi-agent RL and single-agent RL. In our analysis, we leveraged the divide-and-conquer approach to show the convergence of all parameters to the solution of ODE (as in Theorem 1). Toward this end, we first show the convergence of all parameters to the average and then the convergence of the average parameter to the solution of the ODE. Even though the latter part can be proved using the results from the single-agent RL literature, the first part (i.e. the consensus error) has to be carefully addressed. Since in the TD update, all parameters have a $(1+\gamma)$-factor (see Appendix C.2, especially the upper bound on the norm of the matrix $B_{l,k}$), which is different from optimization problems solved by gradient-type approaches, where the parameter doesn't have an expanding factor (or equivalently, the factor is simply 1).
> >
> > > **Your Commment:** 4. For the communication complexity, it is unknown if the proposed method can match batch-based TD methods. It is useful if some lower bound can be established for the proposed methods.
> >
> > **Our Response:** Thanks for your comments, which we fully agree. Under the same sample complexity, the communication complexity of the local TD approach is unknown whether it can match the batch approach since, for both approaches, only the upper bounds have been provided so far. With a reasonable lower bound on sample complexity, it may provide a clearer picture. However, lower bound complexity is an open problem even in the single-agent setting. Deriving a sample complexity lower bound is significant enough to warrant an independent paper in its on right. To the best of our knowledge, all papers on policy evaluations are establishing upper bounds, and no lower bound is known. As a result, all comparisons in the literature are based on upper-bounds (see [G-K]).
> >
> > [G] Srikant R, Ying L. Finite-time error bounds for linear stochastic approximation andtd learning. InConference on Learning Theory 2019 Jun 25 (pp. 2803-2830). PMLR.
> >
> > [H] Xu T, Wang Z, Liang Y. Improving sample complexity bounds for (natural) actor-critic algorithms. Advances in Neural Information Processing Systems. 2020;33:4358-69.
> >
> > [I] Hairi F, Liu J, Lu S. Finite-Time Convergence and Sample Complexity of Multi-Agent Actor-Critic Reinforcement Learning with Average Reward. InInternational Conference on Learning Representations 2021 Sep 29.
> >
> > [J] Chen Z, Zhou Y, Chen RR, Zou S. Sample and communication-efficient decentralized actor-critic algorithms with finite-time analysis. InInternational Conference on Machine Learning 2022 Jun 28 (pp. 3794-3834). PMLR.
> >
> > [K] Qiu S, Yang Z, Ye J, Wang Z. On finite-time convergence of actor-critic algorithm. IEEE Journal on Selected Areas in Information Theory. 2021 May 19;2(2):652-64.
> >
> > > **Your Commment:** 5. It is expected to see that the information of network topology should be a big factor in the final error bound. This is missing. Experiments on other network topologies should be included for illustration.
> >
> > **Our Response:** Thanks for your suggestion. For our theoretical results, we have used Assumption 3, where a quantity $\eta$ is introduced to denote the lower bound of the non-zero entries of the adjacency matrix $A$. This term largely characterizes the network topology and the weights among the edges. Theorem 1 incorporates this term through $\rho$, which is defined as $\rho:=(1+4\beta K)(1-\eta^{N-1})$. Empirically, we have extensively simulated with cooperative navigation tasks using topologies in Appendix Section B.3, where the comparisons of algorithms are illustrated in Figures 9-13. We have also added Figure 8 to show how different network topologies affect our proposed local TD algorithm.
> >
> > **Please continue to see our next response below.**

---

> > > ### Author Response · Authors · 2022-11-19
> > > **Response to Reviewer N76j (Continued)**
> > >
> > > > **Your Commment:** 6. The literature review misses many recent works on MARL policy evaluation problems, which is detrimental to the novelty. A more complete comparison is recommended, e.g., the paper: Taming Communication and Sample Complexities in Decentralized Policy Evaluation for Cooperative Multi-Agent Reinforcement Learning.
> > >
> > > **Our Response:** Thank you for the suggestion and we have added the paper to the related work section. The paper mentioned by the reviewer formulates the policy evaluation as a mean-square projected Bellman error optimization problem, where it is assumed the data is sampled in the steady state, i.e. assuming the mixing time is 0. In contrast, in our paper, we considered both transient and steady-state behaviors. Moreover, an important technical difference between the mentioned paper and our work is that we don't assume certain Lipschitz properties (see Bullet 1 in technical challenges on Page 2 in our paper). By contrast, in the aforementioned paper, the objective function is assumed to be Lipschitz (see Assumption 1(b)).
> > >
> > > > **Your Commment:** 7. Although the agent-drift seems to be unique challenge, the authors haven't highlight how this fails naive application of recent TD analysis. 'Agent drift' phenomenon is interesting. Is there a way to quantify it? For the algorithm analysis, it is important to clarify what is the new development for analyzing the proposed algorithm compared with the mean-path result from Srikant \& Ying, 2019.
> > >
> > > **Our Response:** Thanks for your comments. The vanilla TD algorithm has very little problem in agent drift since it operates consensus communication after every local TD update. However, in our local TD algorithm, the agent drift phenomena will become increasingly obvious as the number of local steps increases. This is captured in the consensus error, which could only be kept small with a carefully chosen local step number k (see right columns of Figure 9-13). As a result, in our proposed algorithm, we need to carefully control the number of local step $K$ so as to maintain a low consensus error.
> > >
> > > When $K$ is larger than the total sample number (or simply infinite), each agent operates local independent learning without any cooperation. As a result, each agent will converge to $w^{i,*}=\Psi^{-1} b^{i}$, where $\Psi$ is defined in Eq. (3) of the main paper and $b^{i}=\mathbb{E}[\phi(s)r^{i}(s,a)]$. As a result, agents fail to learn the global value function.
> > >
> > > In comparison with the paper (Srikant \& Ying, 2019), the mean-path corresponds to centralized learning (suppose there's a central learner that gathers all local information and operates the learning), while our setting is fully decentralized without any central learner.
> > >
> > >
> > > > **Your Commment:** 8. It is important to characterize conditions when the proposed method works and fails.
> > >
> > > **Our Response:** Thanks for your comments. Consensus error in Eq. (7) quantifies how far each parameter deviates from the average parameter. When local step $K$ is too large, the upper bound analysis of the proposed algorithm will fail. In order to avoid this, our theoretical result suggests $K=\Theta(\sqrt{1/\epsilon}\log(1/\epsilon))$.
> > >
> > > > **Your Commment:** 9. What is the specific condition for the inverse in Eq. (4) to exist?
> > >
> > > **Our Response:** Thanks for your question. The condition for the inverse to exist is the Assumption 4. The inverse argument has been provided in [L].
> > >
> > > [L] Tsitsiklis J, Van Roy B. Analysis of temporal-diffference learning with function approximation. Advances in neural information processing systems. 1996;9.
> > >
> > >
> > >
> > > > **Your Commment:** 10. How does the mixing time in Eq. (5) relate to Assumption 1?
> > >
> > > **Our Response:**  The Assumption 1 ensures the unique stationary distribution. Furher, from any initial point (or distribution), it takes $O(\log \frac{1}{\beta}$) iterations for the state distributions to be $\beta$-close to the stationary distribution.

---

### Official Review · Reviewer_YnpC · 2022-11-03

**Confidence:** 3
**Correctness:** 4
**Technical Novelty And Significance:** 3
**Empirical Novelty And Significance:** 3
**Recommendation:** 6

**Clarity, Quality, Novelty And Reproducibility:**

Clarity: The paper is mostly clear in my opinion. The motivation and background are well introduced. The technical challenges are well covered. However, I do hope to learn more about the role of Assumption 1, such as why and how it matters.


Quality: The quality is fine. In terms of theory, the assumptions and theorems are stated without ambiguity. Although I did not check all the proof, the proof looks valid to me. The experiments are extensive enough, and details are included.


Novelty: This paper is novel. It gives the analysis of multiple local TD steps’s communication complexity, which has been an open problem. Technically novelty is also discussed to handle the three coupled challenges.


Reproducibility: The reproducibility seems fine. The experiments details are included. However there is no code uploaded so it is hard to be certain whether the experiments can be reproduced.


**Strength And Weaknesses:**

Strength:

1. Result novelty: this paper conducted a theoretical analysis of the approach of multiple local TD steps in terms of communication complexity, and showed that this approach indeed lowers communication complexity, which has not been shown before. This seems novel to me.

2. Technical novelty: the analysis of this paper has its technical novelty. Although the paper borrows technical tools from other works, it faces coupled challenges due to: underlying structure of TD, dynamic setting (i.e. MDP), and reward heterogeneity across agents.

3. Theoretical results are clearly stated without ambiguity.

4. Extensive empirical results are included which supports the theory.

---
Weakness:
There is no major technical weakness of this paper in my opinion. Still, I think the following is worth mentioning.

1. It is not clear to me why Assumption 1 matters. Specifically, what would happen/why analysis fails if the stationary distribution is not unique?
Is Assumption 1 very practical?
Does it simplify the problem? What might be the extra challenges if the assumption does not hold?

---
Minor comments/Questions:

1. Assumption 4: each agent shares the same value function (i.e. same feature $\phi$ and parameter $w$)? The current statement is confusing. Why not say the global value function $V$ adopts a linear structure?

2. Typo: above section 3.2: definition of V-function: $\gamma^t$ instead of $\gamma$.


**Summary Of The Paper:**

This paper considers the multi-agent reinforcement learning policy evaluation (MARL-PE) problem, where $N$ agents collaborate to evaluate the value function of the global states for a target policy. It focuses on how to analyze the communication cost among agents when using the local temporal-difference (TD) learning method.
First, this paper theoretically shows that executing multiple local TD steps can lower the communication complexity of MARL-PE compared to vanilla consensus-based decentralized TD learning algorithms.
Second, the paper theoretically shows that executing multiple local TD steps has higher communication complexity than the batch approach under same sample complexity condition.
Finally, executing multiple local TD steps is shown to perform similarly to the batch approach in practice under certain settings.


**Summary Of The Review:**

My current recommendation is to accept.

My reasons are the following:
Technical novelty: this paper gives a first answer to an open problem under certain settings
Good quality: the paper is well written: clear presentation, enough motivation. Detailed experiments, etc.

---

> ### Author Response · Authors · 2022-11-19
> **Response to Reviewer YnpC**
>
> We sincerely thank the reviewer's constructive comments and valuable insights, which help improve the quality of our work significantly. We have carefully revised our paper according to your comments and suggestions. Please see our revised submission, where we have highlighted all major changes in **Red color**. Please also see our point-to-point responses as follows:
>
>
>
> > **Your Comment:** 1. It is not clear to me why Assumption 1 matters. Specifically, what would happen/why analysis fails if the stationary distribution is not unique? Is Assumption 1 very practical? Does it simplify the problem? What might be the extra challenges if the assumption does not hold?
>
>
>
> **Our Response:** Thanks for your comments. Please see our response to each of your questions below:
>
> Assumption 1 is a standard assumption in the reinforcement learning policy evaluation literature [A-D] and a very common assumption in stochastic system [E].
>
> [A] Srikant R, Ying L. Finite-time error bounds for linear stochastic approximation andtd learning. InConference on Learning Theory 2019 Jun 25 (pp. 2803-2830). PMLR.
>
> [B] Xu T, Wang Z, Liang Y. Improving sample complexity bounds for (natural) actor-critic algorithms. Advances in Neural Information Processing Systems. 2020;33:4358-69.
>
> [C] Zhang K, Yang Z, Liu H, Zhang T, Basar T. Fully decentralized multi-agent reinforcement learning with networked agents. InInternational Conference on Machine Learning 2018 Jul 3 (pp. 5872-5881). PMLR.
>
> [D] Hairi F, Liu J, Lu S. Finite-Time Convergence and Sample Complexity of Multi-Agent Actor-Critic Reinforcement Learning with Average Reward. InInternational Conference on Learning Representations 2021 Sep 29.
>
> [E] Srikant R, Ying L. Communication networks: an optimization, control, and stochastic networks perspective. Cambridge University Press; 2013 Nov 14.
>
> Assumption 1, i.e. irreducibility and aperiodicity combined together, implies a nice property: the unique existence of the stationary distribution in the limiting behavior. With Assumption 1, we can ensure that the RL system is always well defined.
>
> Let's first look at irreducibility, which means that every state can be reached from every other state, either by one hop or multiple hops (but within a finite number of hops). If irreducibility doesn't hold, it means there are states that can't be reached after a finite number of hops. It further implies that the system has an absorbing subset of irreducible states, which matters in the limiting behavior. In other words, if we are interested in the limiting behavior, the system will only be in those absorbing subsets of irreducible states. So if we only consider this subset of states as the state space, it is irreducible.
>
> Next, the period of a state is the greatest common divider of all possible numbers of hops coming back to the same state. If the period of a state is 1, then the state is aperiodic. Note that if there's a state that is self-transition, then the period is 1. It can be easily shown that, for an irreducible Markov chain, if one state is aperiodic, then all states are aperiodic.
>
> Assumption 1 is very practical in finite-state Markov chain, as we assumed in the paper. Simply put, we can always find an irreducible set of states and often there are states that can self-transit. In this scenario, the Markov chain is irreducible and aperiodic.
> On the other hand, if one considers a non-fully-irreducible Markov chain, then there will occur a finite number of hops of states that relates to *transient* states, which can be considered a noise compared to the Markov chain satisfying Assumption 1.
>
> If one considers a periodic Markov chain, the resulting limiting behavior might *not* be described by any single probability distribution function. For example, consider a two-state (a and b) Markov chain with transition probability$\begin{bmatrix}
> 0 & 1\newline
> 1 & 0
> \end{bmatrix}$, then at a certain time $t$ that is in the steady state, the probability distribution is different from that of at time $t+1$ depending on the initial state (or distribution). That is to say, if the initial state at $t=0$ is a, the distribution is
> $$
> \mathbb{P}(\text{at state a time $t$})= \begin{cases}
>       1 & t \text{ is odd} \newline
>       0 & t \text{ is even}.
>    \end{cases}
> $$However, this is not a stationary distribution, since the distributions at time $t$ and $t+1$ are not the same. In other words, there's no stationary distribution for this Markov chain.
>
> **Please continue to see our next response below.**

---

> > ### Author Response · Authors · 2022-11-19
> > **Response to Reviewer YnpC (Continued)**
> >
> >
> > > **Your Commment:** 2. Assumption 4: each agent shares the same value function (i.e. same feature $\phi$ and parameter $w$)? The current statement is confusing. Why not say the global value function $V$ adopts a linear structure?
> >
> >
> >
> > **Our Response:** Thanks for the suggestion and we reflected this in the paper. Yes, indeed, we try to say the global value function adopts a linear approximation structure and each agent maintains a local copy of the parameter, which can be different from other agents'. We have revised our paper according to your suggestion.
> >
> > > **Your Commment:** 3. Typo: above section 3.2: definition of V-function: $\gamma^{t}$ instead of $\gamma$.
> >
> > **Our Response:** Thanks for pointing this out. We have revised it accordingly.

---

### Decision · Program_Chairs · 2023-01-20

**Decision:**

Reject

**Justification For Why Not Higher Score:**

There is still a gap between the theory and the experiments. The results presented in this paper are not conclusive.

**Justification For Why Not Lower Score:**

N/A

**Metareview: Summary, Strengths And Weaknesses:**

This paper aims to answer the following question: Does local TD-steps work for policy evaluation in the multi-agent setting and how does it compare with batched TD? This paper shows that in theory, local TD has a worse communication complexity than TD with batching, while in experiments they perform similarly.

Strengths:

+ The MARL studied in this paper is an important problem

+ The paper is well written

Weaknesses:

- There is still a gap between the theory and the experiments. According to the experiments, it is very likely that the worse communication complexity for local TD-steps is an artifact of the analysis.

- The dependence on the network topology in the complexity is missing, which again suggests that the analysis is loose.

- The discussion on the related work is not complete.

The main concern is that the results presented in this paper are not conclusive. Even after the author's response, it does not gather sufficient support from the reviewers.